# The STING agonist IMSA101 enhances chimeric antigen receptor T cell function by inducing IL-18 secretion

Ugur Uslu [1,2,8], Lijun Sun[3,8], Sofia Castelli [1,2], Amanda V. Finck [1,2], Charles-Antoine Assenmacher[4], Regina M. Young[1,2], Zhijian J. Chen [5,6,7,9] ✉ & Carl H. June [1,2,9] ✉

As a strategy to improve the therapeutic success of chimeric antigen receptor T cells (CART) directed against solid tumors, we here test the combinatorial use of CART and IMSA101, a newly developed stimulator of interferon genes (STING) agonist. In two syngeneic tumor models, improved overall survival is observed when mice are treated with intratumorally administered IMSA101 in addition to intravenous CART infusion. Transcriptomic analyses of CART isolated from tumors show elevated T cell activation, as well as upregulated cytokine pathway signatures, in particular IL-18, in the combination treatment group. Also, higher levels of IL-18 in serum and tumor are detected with IMSA101 treatment. Consistent with this, the use of IL-18 receptor negative CART impair anti-tumor responses in mice receiving combination treatment. In summary, we find that IMSA101 enhances CART function which is facilitated through STING agonist-induced IL-18 secretion.

While chimeric antigen receptor T cells (CART) have achieved therapeutic success in hematological malignancies[1,2], similar results have not been observed in patients with solid tumors[3]. Reasons for this include insufficient CART trafficking into the tumor, and once entered, impaired anti-tumor efficacy due to an immunosuppressive tumor immune microenvironment (TIME) which causes CART to acquire a dysfunctional or exhausted T cell state[3,4]. To overcome these hurdles, CART therapy in combination with different therapies targeting the TIME are currently under evaluation in preclinical as well as clinical settings[4,5].

In this context, a possible strategy to increase the efficacy of CART therapy could be the combined use of CART with stimulator of interferon genes (STING) agonists. The STING signaling pathway plays a critical role in activating the innate immune defense against cytosolic

DNA associated with microbial infections or damaged cells from tumors[6,7]. In response to cytosolic DNA, cyclic GMP-AMP Synthase (cGAS) generates cyclic GMP-AMP (cGAMP), which binds as an endogenous second messenger to the protein STING, triggering the secretion of proinflammatory type I interferons (IFN) and chemokines[8,9]. STING-activating agents are potentially effective anti-tumor treatments given their ability to induce both innate and adaptive immune responses and to turn immunologically *"cold"* tumors *"hot"*[10-13].

In accordance with this, preclinical reports combining STING agonists with CART were recently published[14,15]. Xu et al. observed that the combined use enhanced tumor control and increased CART trafficking into the tumor and persistence in the TIME[14]. Another group found that the joint action of CART and STING agonist promoted cross priming which counteracted tumor escape mediated by antigen-loss

[1]Center for Cellular Immunotherapies, Department of Pathology and Laboratory Medicine, University of Pennsylvania Perelman School of Medicine, Philadelphia, PA 19104, USA. [2]Parker Institute for Cancer Immunotherapy at University of Pennsylvania, Philadelphia, PA 19104, USA. [3]ImmuneSensor Therapeutics, Dallas, TX 75235, USA. [4]Comparative Pathology Core, Department of Pathobiology, School of Veterinary Medicine, University of Pennsylvania, Philadelphia, PA 19104, USA. [5]Department of Molecular Biology, University of Texas Southwestern Medical Center, Dallas, TX 75390, USA. [6]Center for Inflammation Research, University of Texas Southwestern Medical Center, Dallas, TX 75390, USA. [7]Howard Hughes Medical Institute, 4000 Jones Bridge Road, Chevy Chase MD20815, USA. [8]These authors contributed equally: Ugur Uslu, Lijun Sun. [9]These authors jointly supervised this work: Zhijian J. Chen, Carl H. June. ✉e-mail: zhijian.chen@utsouthwestern.edu; cjune@upenn.edu

variants[15]. However, both studies used either 2'3'cGAMP and/or DMXAA as agonists[14,15]. Although promising results were seen in animal models, DMXAA treatment showed no efficacy in patients, and further analyses revealed that mouse, but not human STING binds and signals in response to DMXAA[16,17]. While 2,3-cGAMP targets human STING, its poor stability in serum makes it less effective in promoting immune response and infiltration of $CD8^+$ T cells into tumors[18]. These limitations have prompted the development of synthetically enhanced cGAMP analogues such as IMSA101, which is under clinical development by ImmuneSensor Therapeutics[19]. In particular, IMSA101 has demonstrated a favorable safety profile in a Phase I clinical trial. Currently, Phase II trials are underway to evaluate the combination of IMSA101 with immune checkpoint inhibitors and radiotherapy (NCT04020185, NCT05846646, NCT05846659).

Herein we show that IMSA101 displays enhanced stability in human serum and improved in vivo anti-tumor efficacy compared to cGAMP. Futher, we demonstrate superior anti-tumor efficacy and survival using the combination of intratumorally (i.t.) administered IMSA101 and intravenously (i.v.) infused CART in syngeneic mouse models of cancer. Mechanistically, we establish that IMSA101 modulates the TIME by increasing infiltration of CART and other immune cells into the tumor, and by inducing a pro-inflammatory cytokine milieu. In addition, we demonstrate that IMSA101-treatment increases IL-18 secretion which enhances CART anti-tumor efficacy. Thus, our study provides mechanistic insights into how STING agonists improve CART function and supports the potential clinical advancement of CART and IMSA101 combination therapy.

## Results

### The cGAMP analogue IMSA101 shows superior serum stability and increased in vivo anti-tumor efficacy

To establish the rationale for combining the cGAMP analogue IMSA101 and CART to improve therapeutic response in solid tumors, we first demonstrated the immunostimulatory activity, superior serum stability, and improved in vivo anti-tumor activity of IMSA101 over conventional cGAMP (Fig. 1).

When STING agonists were added to THP1-ISG-luc, a human monocyte cell line with a luciferase reporter under the control of a promoter with ISREs (interferon stimulated response elements), IMSA101 induced an IFN response with a half maximal effective concentration ($EC_{50}$) of 8μM, which was similar to the potency of cGAMP (Fig. 1a). In the presence of perfringolysin O (PFO), a pore-forming agent that allows free entry of small molecules through the plasma membrane, IMSA101 stimulated an IFN response with an $EC_{50}$ value of 20–30 nM, which was again similar to that of cGAMP (Fig. 1b). To confirm that IMSA101 specifically induces IFN response through the

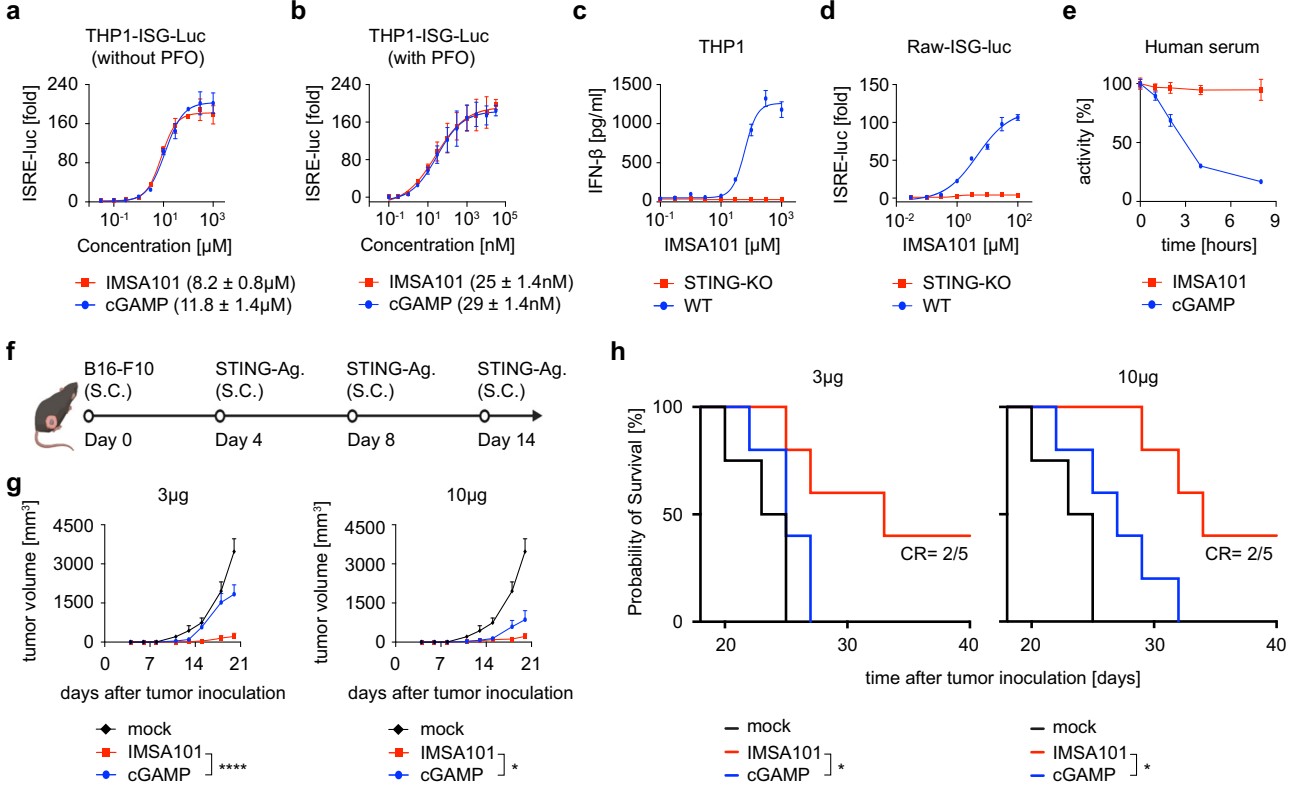

**Fig. 1 | IMSA101 shows superior serum stability and in vivo anti-tumor efficacy. a** THP1-ISG-luc cells were incubated with serial concentrations of IMSA101 or cGAMP for 16 h, followed by measurement of luciferase activity. **b** In addition to the procedure as described in (**a**), 25 ng/ml of PFO was included in the cell media. **c** Wildtype (WT) or STING knockout (STING-KO) THP1 cells were stimulated with indicated concentrations of IMSA101 for 16 h, followed by measurement of IFN-β in the media. **d** WT or STING-KO Raw-ISG-luc cells were incubated with indicated concentrations of IMSA101 for 16 h and luciferase activity in media was quantified. For (**a**–**d**) mean values ± SD of three biologically independent samples are shown. EC50 values were derived from curves fitted to four-parameter Hill equation. **e** IMSA101 or cGAMP was incubated with human serum for a total of eight hours. The reaction was terminated by 5 min boiling, and supernatant after centrifugation was incubated with THP1-ISG-luc cells to quantify bioactivity. Mean values ± SD of three biologically independent samples are shown. **f** Schematic of experimental design for in vivo comparison of IMSA101 to cGAMP. Icons of this figure created with BioRender.com. **g** Changes in tumor volume over time at indicated STING-agonist doses ($n = 5$/cohort for cGMAP and IMSA101 groups; $n = 4$ for mock group). Two-way ANOVA with Sidak's test was used for pairwise multiple comparisons. *$P \le 0.05$; ****$P \le 0.0001$. Data are presented as mean values +/− SD. **h** Kaplan-Meier survival curve at indicated STING-agonist doses ($n = 5$/cohort for cGMAP and IMSA101 groups; $n = 4$ for mock group). Statistical significance was calculated using the log-rank Mantel-Cox test. *$P \le 0.05$. CR, complete remission. Source data and exact p-values are provided as a Source Data file.

STING pathway, IMSA101 was co-incubated with wild-type (WT) or STING knock-out (STING-KO) THP1-ISC-luc (Fig. 1c) as well as Raw 264.7, a murine macrophage ISG-luc reporter cell line (Fig. 1d). While co-incubation of IMSA101 with both WT reporter cell lines stimulated an IFN response, deletion of the STING gene in these cells (STING-KO) completely abolished IFN response to IMSA101 (Fig. 1c, d).

Since poor stability has limited the clinical efficacy of cGAMP targeting human STING, we evaluated the stability of IMSA101 compared to conventional cGAMP in human serum (Fig. 1e). Incubation of IMSA101 in human serum showed robust stability and did not result in reduced activity over the duration of the observed period, while cGAMP rapidly lost its activity over time (Fig. 1e).

Next, we compared the in vivo antitumor activity of IMSA101 with that of cGAMP in a syngeneic mouse tumor model using B16-F10, a murine melanoma cell line (Fig. 1f). Mice were subcutaneously (s.c.) inoculated with tumor cells into their right flank and then received s.c. injections of IMSA101, cGAMP or PBS (mock) (3 µg or 10 µg). Injections with PBS were used as control. Treatment of mice with cGAMP slowed tumor growth in a dose-dependent manner (Fig. 1g), and the 10µg resulted in a modest improvement of overall survival (Fig. 1h). In contrast, both doses of IMSA101 controlled tumor growth and significantly improved overall survival (Fig. 1g, h), with 40% of mice in both IMSA101 cohorts exhibited complete remission of flank tumors (Fig. 1h) without tumor recurrence over the observed period of 6 months.

Taken together, these data demostrate that IMSA101 spefically induces IFN secretion through the STING-pathway and displays superior serum stability over conventional cGAMP. Further, IMSA101 shows improved in vivo anti-tumor efficacy in syngeneic mice, supporting its ongoing clinical translation and providing rationale for a CART combinatorial treatment approach.

## IMSA101 treatment improves CART efficacy against solid tumors

To determine whether i.t. administration of the STING agonist IMSA101 enhances CART efficacy in solid tumors, two syngeneic flank tumor models using immunocompetent CD45.2$^+$ C57BL/6 mice were established (Fig. 2a). Mice were either implanted with murine mesothelin-expressing PDA7940b, an immunologically *"cold"* murine pancreatic ductal adenocarcinoma (PDA) cell line derived from KPC (Kras$^{LSL.G12D/+}$p53$^{R172H/+}$) mice, or with B16-F10 transduced with human CD19 (B16-huCD19) (Supplementary Fig. 1a). Mouse T cells were prepared from the spleen of CD45.1$^+$ B6.SJL-Ptprc$^a$ Pepc$^b$/BoyJ mice, and either transduced with a murine mesothelin-specific CAR construct to be used in the PDA7940b model, or with an anti-human CD19-specific CAR construct to be used in the B16-huCD19 animal model. CAR transduction rates were analyzed via cell surface staining and flow cytometry before infusion into mice. Approximately 75% of T cells expressed the mouse mesothelin-specific and 50% of T cells expressed the human CD19-specific CAR construct (Supplementary Fig. 1b).

In addition to the syngeneic models, a xenograft flank tumor model was used which, unlike syngeneic mice, lacks an immunosuppressive TIME and allows us to analyze direct effects of IMSA101 on human CART. To establish the xenograft model, human PDA AsPC-1 cells were injected s.c. into the right flank of immunodeficient NOD/scid/IL2ry$^{-/-}$ (NSG) mice (Supplementary Fig. 2a). AsPC-1 cells endogenously express human mesothelin (Supplementary Fig. 1c) which served as the target antigen for the CAR construct. T cells from two different healthy human donors were used (ND517 and ND569) to confirm that observed effects were donor independent. Approximately 52% and 58% of donors T cells ND517 and ND569, respectively, expressed the human mesothelin-specific CAR construct upon infusion into mice (Supplementary Fig. 1d).

Significantly improved overall survival was observed in mice receiving combination treatment in both syngeneic animal models when compared to mice receiving IMSA101 alone or CART alone (Fig. 2b). Three out of ten mice receiving IMSA101 + CART showed complete remission of the tumor in the PDA7940b model, while all mice (ten out of ten) cleared tumors in the B16-huCD19 model (Fig. 2c). Also, four out of ten mice receiving IMSA101 alone showed complete remission in the B16-huCD19 animal model, while all mice in this cohort in the PDA7940b model progressed and required sacrifice (Fig. 2c). Untreated mice, as well as mice receiving CART alone showed rapid tumor progression over time which required sacrifice in both cohorts (Fig. 2c). None of the mice receiving combination treatment showed clinical signs of distress, or severe weight loss over time (Supplementary Fig. 3a). Some mice in the untreated or CART alone groups showed weight loss over time, which correlated with tumor progression (Supplementary Fig. 3a).

In contrast, minimal effects of IMSA101 on CART were observed in immunodeficient mice (Supplementary Fig. 2). A modest overall survival benefit was seen between IMSA101 + CART and IMSA101-alone cohorts, but not between IMSA101 + CART and CART-alone cohorts when donor T cells ND517 were used (Supplementary Fig. 2b). No survival differences between cohorts were observed with ND569 donor T cells (Supplementary Fig. 2b). One out of 20 mice treated with IMSA101 + CART showed complete remission, while the rest of the mice in this cohort, and all mice in other cohorts showed tumor progression over time (Supplementary Fig. 2c).

To test whether IMSA101 stimulates endogenous immunity, a total of 14 mice with complete remission in the immunocompetent B16-huCD19 animal model (ten out of ten mice of the IMSA1010 + CART group as well as four out of ten mice of the IMSA101-alone group, see Fig. 2c) were re-challenged with B16-F10 tumor cells (not expressing human CD19) into the contralateral flank 60 days after treatment and 40 days free of palpable tumors (Fig. 2d). Treatment-naïve mice were implanted with B16-F10 tumor cells as control. Improved overall survival was seen in re-challenged mice when compared to treatment-naïve mice (Fig. 2e and Supplementary Fig. 4a). Two out of 10 re-challenged mice of the IMSA101 + CART group, and two out of four re-challenged mice of the IMSA101-alone group exhibited no tumor engraftment and the remaining mice showed delayed engraftment compared to the treatment-naïve mice (Fig. 2f and Supplementary Fig. 4b). Weight loss over time was primarily observed in some treatment-naïve mice which correlated with tumor progression (Supplementary Fig. 4c).

Next, we analyzed whether IMSA101 induced abscopal effects. Immunocompetent C57BL/6 mice were implanted with B16-huCD19 tumor cells into both flanks (Fig. 2g). Additionally, CART were administered i.v., but only the right flank tumor received i.t. treatment with IMSA101. Significantly improved overall survival was observed in IMSA101 + CART treated mice when compared to mice receiving IMSA101 alone or CART alone (Fig. 2h). Caliper measurement of the treated right flank tumor showed complete remission in all ten mice of the IMSA101 + CART group (Fig. 2i). Four of these mice also showed complete remission or no tumor engraftment in the contralateral left flank tumor ( = untreated), while the other six mice of this group required sacrifice following progression of the left flank tumor (Fig. 2i). In the IMSA101-alone group three out of ten mice showed complete remission of the treated right flank tumor, while all mice still required sacrifice due to progression of the untreated left flank tumor (Fig. 2i). All mice receiving CART alone or no treatment showed tumor progression on both flanks (Fig. 2i). Body weight loss over time was observed in some mice throughout all cohorts and this correlated with tumor progression (Supplementary Fig. 3b).

In summary, IMSA101 treatment significantly increased CART efficacy and improved overall survival in two different syngeneic models. Further, potential epitope spreading and abscopal effects

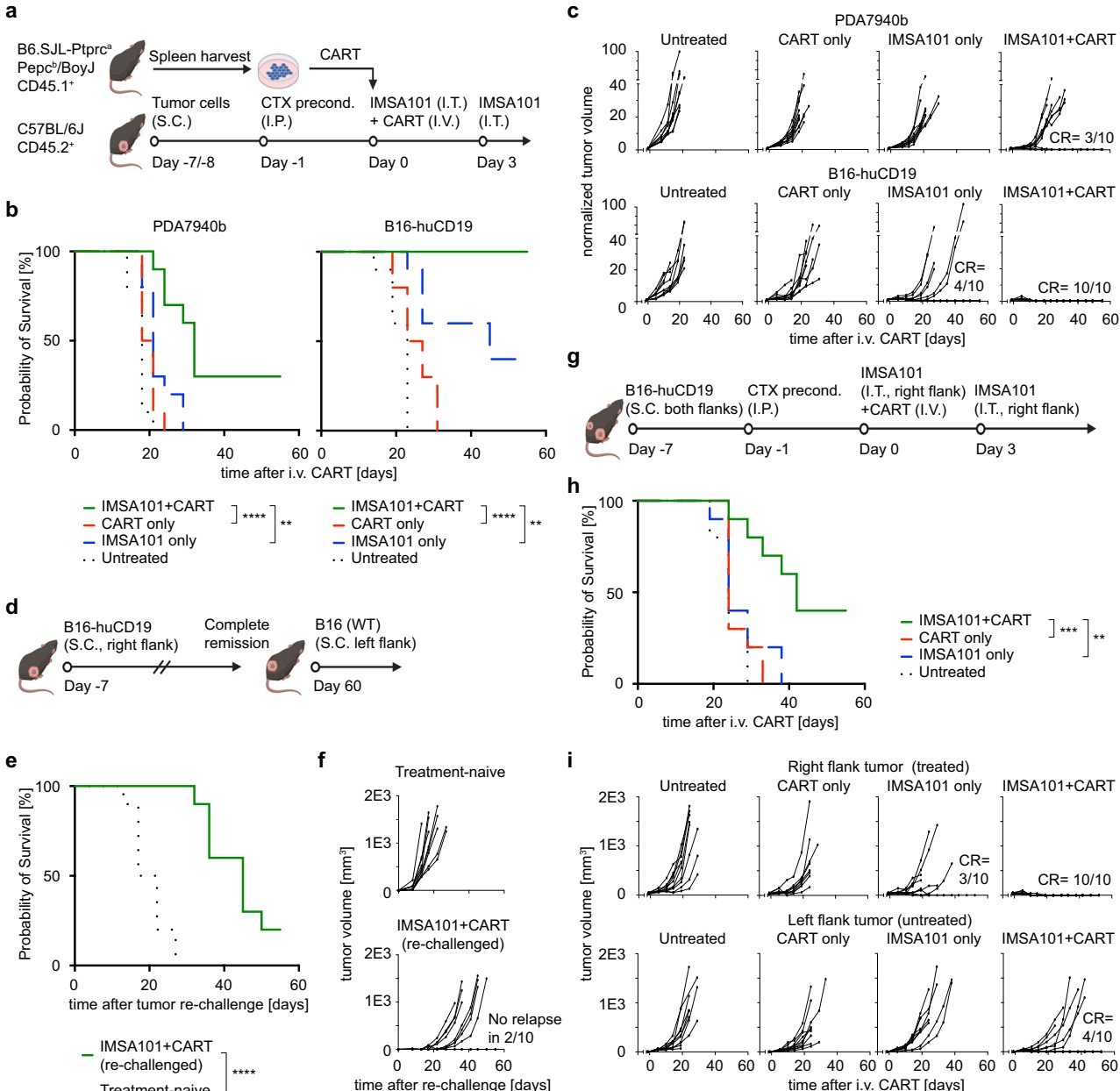

**Fig. 2 | Combining i.t. IMSA101 with i.v. CART increases anti-tumor response, induces memory T cell formation, and promotes abscopal effects in syngeneic animal models. a** Schematic of experimental design for testing combinatorial treatment approach in flank tumor models. Icons of this figure created with BioRender.com. **b** Kaplan-Meier survival curve (*n* = 10/cohort). Statistical significance was calculated using the log-rank Mantel-Cox test. **P ≤ 0.01; ****P ≤ 0.0001. **c** Changes in tumor volume over time. CR, complete remission. **d** Schematic of experimental design for tumor re-challenge study. Icons of this

figure created with BioRender.com. **e** Kaplan-Meier survival curve (*n* = 10/cohort). Statistical significance was calculated using the log-rank Mantel-Cox test. ****P ≤ 0.0001. **f** Changes in tumor volume over time. **g** Schematic of experimental design for dual-flank tumor model. Icons of this figure created with BioRender.com. **h** Kaplan-Meier survival curve (*n* = 10/cohort). Statistical significance was calculated using the log-rank Mantel-Cox test. **P ≤ 0.01; ***P ≤ 0.001. **i** Changes in tumor volume over time. CR, complete remission. Source data and exact p-values are provided as a Source Data file.

were observed in tumor re-challenged mice and in a dual-flank tumor model, respectively. As expected, minimal effects were seen in immunodeficient mice, supporting the hypothesis that STING agonist-induced modulation of the TIME and induction of a pro-inflammatory cytokine milieu enhances antitumor CART-immunity.

## IMSA101-treatment improves CART and immune cell tumor infiltration

To characterize changes in the TIME following IMSA101 treatment, single cell suspension of tumor cells was analyzed by flow cytometry. Pathological assessment, immunohistochemistry (IHC), and RNA

in situ hybridization (RNA-ISH) were performed at depicted times on bulk tumor tissue of syngeneic (Fig. 3 and Supplementary Fig. 5) and xenograft mice (Supplementary Fig. 6). Additionally, peripheral blood and single cell suspension of the spleen were analyzed by flow cytometry to determine effects of IMSA101 on extra-tumoral CART (Supplementary Figs. 5, 6).

Flow cytometry analyses revealed significantly increased levels of infused CD3$^+$CD45.1$^+$ cells in PDA7940b tumors from mice receiving the combination of IMSA101 and CART compared to CART alone (Fig. 3b and Supplementary Fig. 7a). In the B16-huCD19 model, a trend towards more intra-tumoral CD3$^+$CD45.1$^+$ cells were

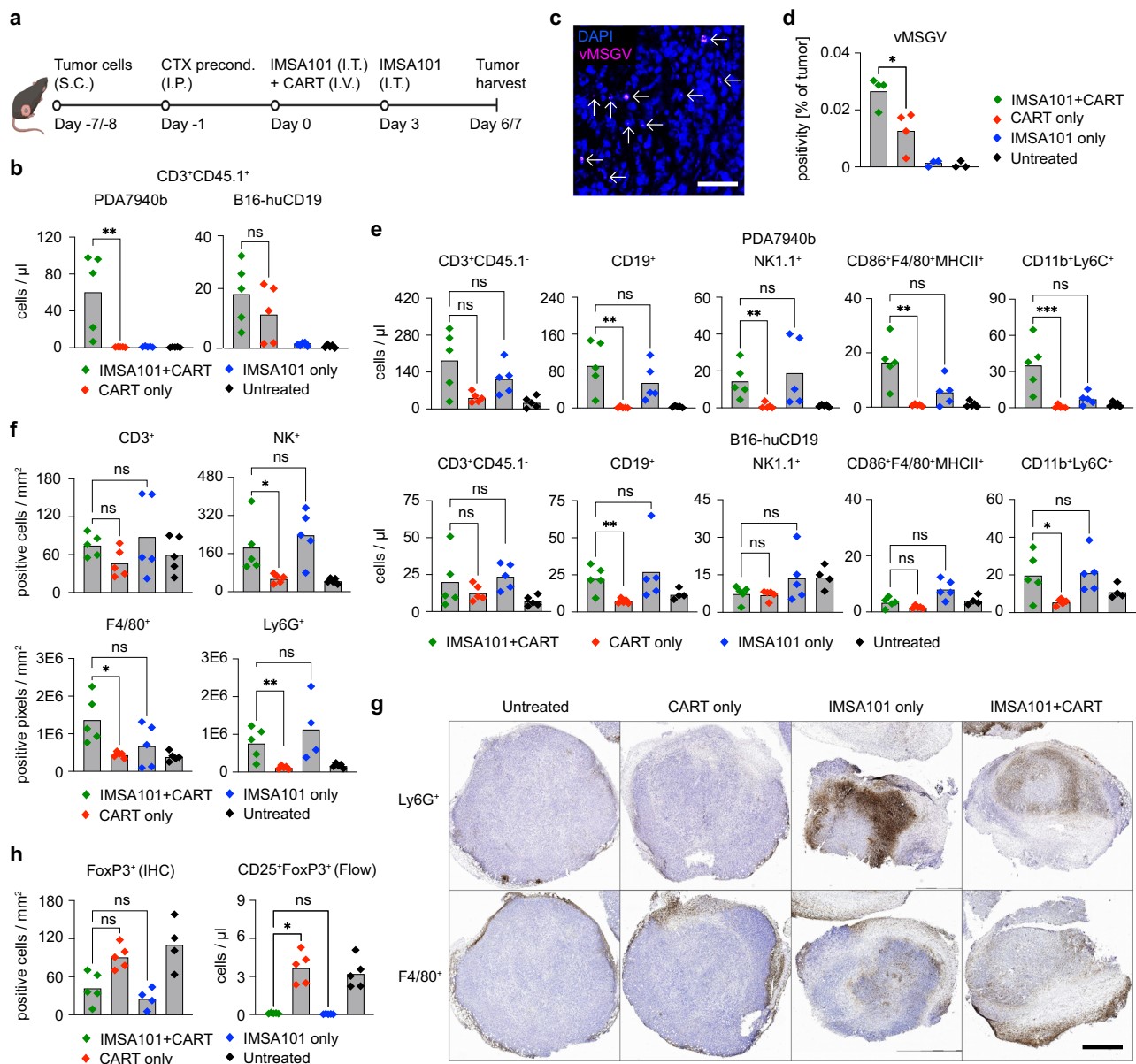

**Fig. 3 | i.t. IMSA101 supports CART and immune cell trafficking into the tumor.**
**a** Schematic of experimental design to analyze changes in tumor immune micro-environment (TIME) following combinatorial treatment. Immunohistochemistry (IHC) and RNA in situ Hybridization (RNA-ISH) was performed from tumors at day 6 after treatment, and flow cytometry analyses from tumors at day 7 after treatment in separate experiments. Icons of this figure created with BioRender.com.
**b** CD3+CD45.1+ cell count ( = infused T cells) analyzed via flow cytometry from tumor single cell suspension (*n* = 5/cohort). Average and individual values are shown. Two-sided Mann-Whitney U Test was used for statistical analysis. ns (non-significant) *P* > 0.05; **P ≤ 0.01. **c**, **d** RNA-ISH of formalin-fixed, paraffin-embedded PDA7940b tumor tissues to detect vMSGV RNA ( = CART). **c** Shows representative RNA-ISH photomicrograph (40x; scale bar equal to 40μm; arrows mark detection of vMSGV RNA), and (**d**) shows vMSGV RNA-positive area count within tumor samples (*n* = 3/cohort for Untreated and IMSA101 only groups; *n* = 4/cohort for IMSA101 + CART and CART only groups). Average and individual values are shown. Two-sided Mann-Whitney U Test was used for statistical analysis. *P ≤ 0.05. **e** Count of different cell populations as indicated, analyzed via flow cytometry from tumor single cell suspensions (*n* = 5/cohort). Average and individual values are shown. Kruskal-Wallis one-way analysis of variance was used for statistical analysis. ns (nonsignificant) *P* > 0.05; *P ≤ 0.05; **P ≤ 0.01; ***P ≤ 0.001. **f**, **g** IHC staining of formalin-fixed, paraffin-embedded PDA7940b tumor tissues. **f** Shows quantification of different cell populations in tumor tissues (*n* = 5/cohort; for Ly6G+ IMSA101 only group *n* = 4). Average and individual values are shown. Kruskal-Wallis one-way analysis of variance was used for statistical analysis. ns (nonsignificant) *P* > 0.05; *P ≤ 0.05; **P ≤ 0.01. **g** Shows representative IHC-photomicrographs (2x) for Ly6G+ and F4/80+ cells (in brown). Scale bar equal to 1 mm. **h** Count of FoxP3-positive cells in IHC from bulk tumor tissue and cell surface staining and flow cytometry from tumor single cell suspension (*n* = 5/cohort; for FoxP3 (IHC) IMSA101 only and Untreated groups *n* = 4/cohort). Average and individual values are shown. Kruskal-Wallis one-way analysis of variance was used for statistical analysis. ns (non-significant) *P* > 0.05; *P ≤ 0.05. Source data and exact *p*-values are provided as a Source Data file.

observed in the combination treatment group (Fig. 3b). To confirm that CART were enriched in tumors treated with IMSA101, we employed RNA-ISH to visualize CAR transcripts in individual cells of formalin-fixed, paraffin-embedded PDA7940b tumor tissues. These assays verified that levels of CART were enriched in tumors in the combination treatment group compared to the CART-alone group (Fig. 3c, d).

In both PDA7940b and B16-huCD19 tumors, significantly more CD19+ cells ( = B cells), and CD11b+Ly6C+ cells ( = Myeloid-derived suppressor cells, MDSCs) were detected in the combination treatment

group when compared to the CART-alone group, while no significant differences were observed between the combination and the IMSA101-alone groups (Fig. 3e). Additionally, in PDA7940b tumors significantly more NK1.1+ cells (= NK cells) and CD86+F4/80+MHCII+ cells (= M1-macrophages) were observed in the combination treatment group when compared to the CART-alone group (Fig. 3e and Supplementary Fig. 5b). Low levels of CD86+F4/80+MHCII-CD206+ cells (= M2-macrophages) were detected in tumors throughout all cohorts and no statistical differences in the number of M2 macrophages were observed between the groups (Supplementary Fig. 5b).

These results were confirmed by pathological assessment and IHC of bulk PDA7940b tumor tissues. Significantly more NK+ cells (= NK cells), F4/80+ cells (= macrophages), and Ly6G+ cells (= neutrophils) were detected in tumors of IMSA101 + CART treated mice when compared to CART-alone groups (Fig. 3f). No significant differences were detected in these cell types between the combination and IMSA101-alone groups (Fig. 3f). Representative IHC images confirmed tumor infiltration with Ly6G+ (= neutrophils) and F4/80+ cells (= macrophages) in the combination treatment as well as IMSA101-only groups, while immune cells were primarily detected at the tumor borders without notable infiltration in untreated and CART-alone cohorts (Fig. 3g). In contrast, less FoxP3+ cells (T_regs) were observed in flow cytometric analyses of single cell suspension from tumors in IMSA101 + CART as well as IMSA101-alone cohorts (Fig. 3h and Supplementary Fig. 7b).

In H&E staining and pathological assessment of tumor tissues, more signs of necrosis and inflammation were observed in tumors of IMSA + CART and IMSA101-alone groups, while fibrosis was not seen in any of the tumors (Supplementary Fig. 5c–e). Also, no differences between cohorts were observed in CD3+CD45.1+ cell counts (= infused T cells) as well as CD3+CD45.1- cell counts (= endogenous T cells) in peripheral blood (Supplementary Fig. 5f) or in spleen (Supplementary Fig. 5g) of treated immunocompetent mice. In addition, no differences of other cell types as indicated were observed in the spleen between cohorts (Supplementary Fig. 5h).

As expected, no enrichment of infused human T cells was observed in the immunodeficient NSG/AsPC-1 tumor model when the combination treatment group was compared to the CART-alone group (Supplementary Fig. 6b–d). Further pathological assessment and H&E staining of tumor tissue in this model revealed no differences in severity of necrosis, inflammation, or fibrosis between cohorts (Supplementary Fig. S6e, f).

In summary, an increased number of CART as well as other immune cells were detected in tumor of IMSA101 + CART-treated syngeneic mice, indicating that IMSA101 remodels the TIME, supporting trafficking of CART and other immune cells into the tumor.

## IMSA101 treatment induces intra-tumoral CART activation and increased expression of immunoregulatory and/or pro-inflammatory pathways

To gain insight into how IMSA101 impacts the phenotype of intratumoral T cells, both CD3+CD45.1+ cells (= infused T cells) and CD3+CD45.1- cells (= endogenous T cells), we analyzed single cell suspensions of tumors by cell surface staining and flow cytometry (Fig. 4a–e and Supplementary Fig. 8, 9). Further, RNA signatures of intratumoral CD3+CD45.1+ cells following combination treatment were analyzed and compared to CART-alone treatment group (Fig. 4f–i).

In flow cytometry analyses, significantly more intratumoral CD8-positive CD3+CD45.1+ cells could be detected in the combination treatment group when compared to CART-alone group. In contrast, levels of CD4-positive CD3+CD45.1+ were significantly higher in the CART-alone compared to the combination treatment group (Fig. 4b and Supplementary fig. 8a), resulting in a significant difference in intratumoral CD4/CD8 cell ratio between IMSA101 + CART and CART-alone cohorts (Fig. 4c). Significant differences in

CD4/CD8 cell ratio were also observed in intratumoral CD3+CD45.1- cells between IMSA101 + CART and CART-alone cohorts, but not between combination treatment and IMSA101-alone groups (Supplementary Fig. 9a).

More intratumoral CD4-positive and CD8-positive CD3+CD45.1+ cells expressed the activation marker PD-1 in tumors receiving IMSA101 + CART when compared to CART-only in both animal models (Fig. 4d). Further, significantly more CD4-positive or CD8-positive CD3+CD45.1- cells expressed PD-1 in the combination treatment group when compared to the CART-alone group in the PDA7940b or B16-huCD19 tumor models, respectively (Supplementary Fig. 9b). Also, significantly more intratumoral T cells, either infused or endogenous, showed CD44-CD62L- effector cell phenotype in the combination group compared to the CART-alone group (Fig. 4e and Supplementary Fig. 9c). No differences were observed between groups in the frequency of dead intratumoral CD3+CD45.1+ cells in either syngeneic model (Supplementary Figs. 8b and 9d). In contrast, a higher frequency of dead CD3+CD45.1- cells was seen between the combination treatment group when compared to the CART-alone group in the PDA7940b tumor model (Supplementary Fig. 9e). No differences in the frequency of dead CD3+CD45.1- cells between cohorts were observed in the B16-huCD19 tumor model (Supplementary Fig. 9e).

Next, we compared the RNA signatures of intratumoral CD3+CD45.1+ cells (= infused T cells) harvested from PDA7940b tumors six days following combination and CART-alone treatment (Fig. 4f). Gene expression and pathway analyses were performed using the NanoString nCounter platform. The nCounter® Immune Exhaustion Panel was used which includes 785 genes across 47 pathways involved in immune activation, immune suppression, immune status, immune checkpoint, epigenetics, and metabolism, and microenvironment.

Analyses of differentially expressed genes (DEGs) in intratumoral CD3+CD45.1+ cells of the combination treatment group revealed that 212 genes were up-regulated, while 10 genes were down-regulated when compared to intratumoral cells of the CART-alone group (Fig. 4g and Supplementary Data 1). Genes with known links to effector T cells, including Eomes, Prf1, Gzmg, Gzmm, Gzmn, and Cxcr3, as well as genes with known links to T cell activation, proliferation, and/or pro-inflammatory functions, including Pdcd1, IL-2, IL-4, IL-9, Cd28, and Icos were significantly upregulated (Fig. 4g volcano plot, right side and Supplementary Data 1). Only a few genes were downregulated, including IL-1a and IL12a, which can be linked to proinflammatory responses (Fig. 4g volcano plot, left side and Supplementary Data 1).

Further, we observed significant upregulation of genes associated with a published T cell activation signature[20], such as Ifna1, Pdcd1, IL-2, Tigit, Tnfrsf4, Gzmb, Xcl1, Ctla4, and Ccl4, in the IMSA101 + CART combination cohort compared to the CART-alone group (Fig. 4h).

Gene Set Variance Analysis (GSVA) was performed in the next step to identify pathways up- or down-regulated in the IMSA101 + CART combination group when compared to the CART-alone group (Fig. 4i and Supplementary Data 2). All significantly upregulated pathways were linked to immunoregulatory and/or pro-inflammatory functions. IL-18 was the most significantly upregulated pathway, followed by IL-21, IL-9, IL-2, IL15, IL-6, and IL-23 pathways (Fig. 4i and Supplementary Data 2). In contrast, TGFβ-receptor and IL-10 pathways, with known links to anti-inflammatory responses, as well as IL-1 and IL-36 with links to anti- and pro-inflammatory responses were significantly downregulated (Fig. 4i and Supplementary Data 2).

Taken together, flow cytometry and transcriptome analyses confirmed that local administration of IMSA101 improves CART function and induces T cell activation and pro-inflammatory cytokine pathways, with IL-18 being the most significantly upregulated pathway in our analyses.

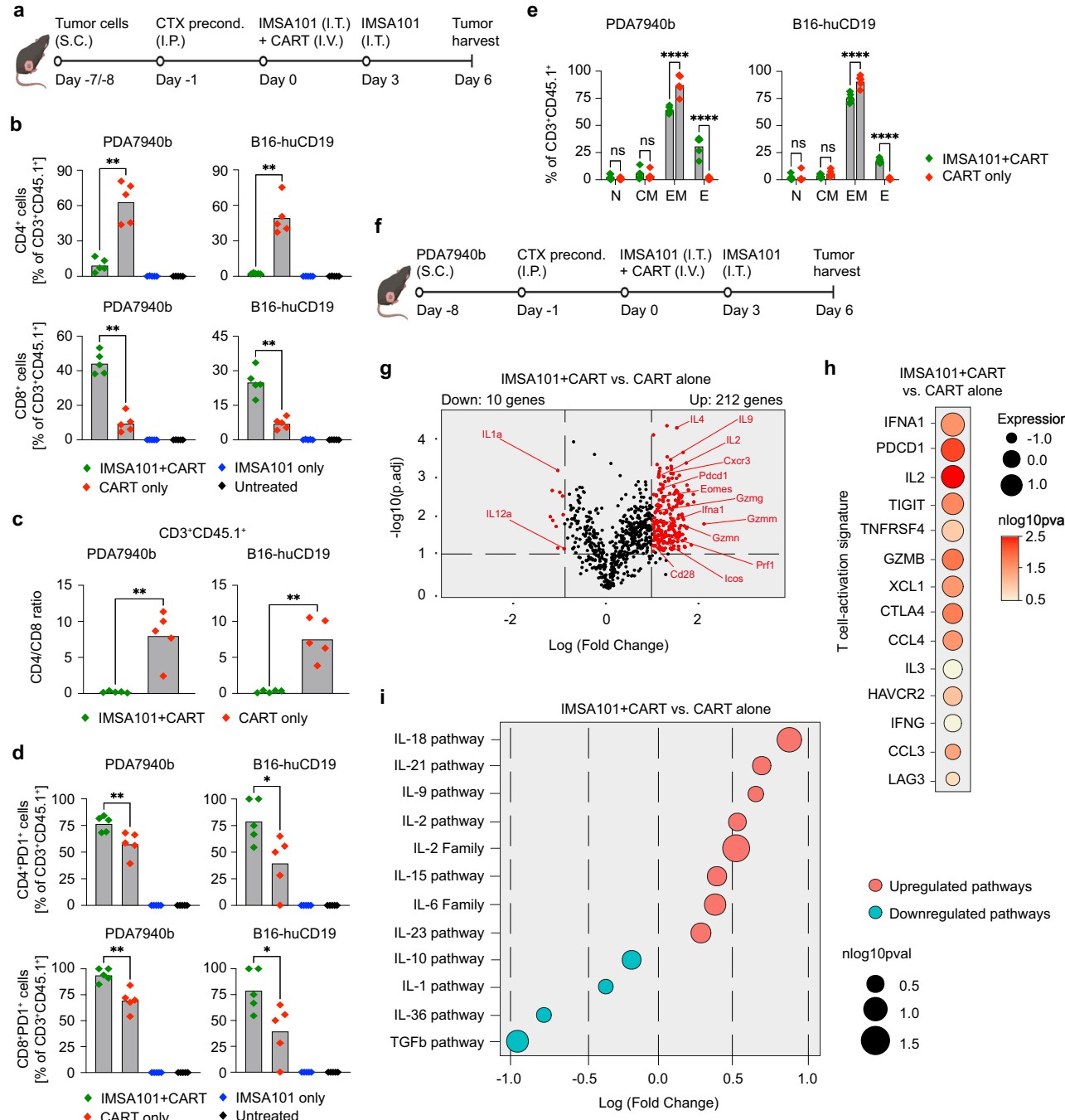

**Fig. 4 | i.t. IMSA101 induces intratumoral CART activation and upregulation of IL-18 and other immune stimulatory pathways. a** Schematic of experimental design to analyze changes of CART and endogenous T cells in tumor microenvironment (TIME) following combinatorial treatment. Flow cytometry analyses was performed from tumors at day 6 after treatment in a single experiment. Icons of this figure created with BioRender.com. **b** CD4/CD8-positive cell count, **c** CD4/CD8 ratio, and **d** rate of PD1-expressing CD3+CD45.1+ cells ( = infused T cells) analyzed via flow cytometry from tumor single cell suspension (*n* = 5/cohort). Average and individual values are shown. Two-sided Mann-Whitney U Test was used for statistical analysis. *$P \leq 0.05$; **$P \leq 0.01$. **e** Phenotype of intratumoral CD3+CD45.1+ cells ( = infused T cells; *n* = 5/cohort). N=naïve (CD44−CD62L+); CM=central memory (CD44+CD62L+); EM=effector memory (CD44+CD62L−); E=effector (CD44−CD62L−). Two-way ANOVA with Sidak's test was used for pairwise multiple comparisions. ns (nonsignificant) *P* > 0.05; ****$P \leq 0.0001$. **f** Schematic of experimental design to

analyze changes on transcriptomic levels of intratumoral CART of IMSA101 + CART group compared to CART-alone group, which was performed from tumors at day 6 after treatment. Tumors from a total of 10 mice (IMSA101 + CART group) or 15 mice (CART-only group) were pooled for isolation of intratumoral CART of IMSA + CART or CART-alone cohorts, respectively. Icons of this figure created with BioRender.com. **g** Volcano plot shows overview of differentially expressed genes (DEGs) of all analyzed 785 genes. Red dots indicate significantly up- or downregulated genes with negative log of adjusted *p*-value (nlog10pval) >1 and log fold change (LFC) >1. t-test, two-sided. **h** Bubble plot shows expression level of T cell activation-associated genes as depicted. t-test, two-sided. **i** Bubble plot shows up- or downregulated pathways as depicted using Gene Set Variance Analysis (GSVA). t-test, two-sided. Source data and exact *p*-values are provided as a Source Data file, as well as Supplementary Data files 1 and 2.

## IMSA101 induces secretion of IL-18 and other pro-inflammatory cytokines

Given the previously reported transcriptomics data, we next sought to analyze cytokine levels in mouse serum and mouse tumors. Mice were sacrificed at day six or seven after treatment, blood was collected by cardiac puncture, and cytokine profiles from serum were analyzed by Luminex assay (Fig. 5a). In addition, mouse tumors were collected and cytokine levels from tumor cell lysates were evaluated. An Enzyme-linked immunosorbent assay (ELISA) was performed to measure levels of IL-18 binding protein (IL-18BP) and IL-18 from mouse serum and mouse tumors, respectively. Further, bulk tumor tissues were formalin-fixed and paraffin embedded for subsequent RNA-ISH experiments.

In the PDA7940b tumor model, significantly more IL-18, IFNγ, and IL-2 was detected in mouse serum from the combination treatment group compared to CART alone (Fig. 5b, c and Supplementary Fig. 10a). Further, significantly more IL-5, IL-13, and TNF was observed in serum of the IMSA101 + CART group compared to CART-alone and IMSA101-alone cohorts (Fig. 5b, c and Supplementary Fig. 10a). In the B16-huCD19 tumor model, significantly more IL-18 and IL-23, as well as a trend towards more IL-22 and IFNγ was detected in serum of mice receiving combination treatment when compared to mice receiving CART alone (Fig. 5b, c and Supplementary Fig. 10b). No statistically significant differences were seen in IL-18BP levels in mouse serum among analyzed treatment cohorts (Supplementary Fig. 10c). IL-18BP has been shown to antagonize the antitumor effect of IL-18[21]. Thus, the increase of IL-18 without concomitant increase of IL-18BP in the IMSA101 plus CART group bodes well for the superior efficacy of the combination therapy.

Next, we measured cytokine levels from PDA7940b tumors. Significantly higher levels of IL-18, IL-22, IL23, GM-CSF, IFNγ, TNF, IL-9, Il-1β, IL-2, and IL-5 were detected in tumors of mice receiving combination treatment when compared to mice receiving CART alone (Fig. 5d and Supplementary Fig. 11a). Further, significantly lower IL-10 levels were observed in the combination treatment group when compared to the CART-only group (Fig. 5d and Supplementary Fig. 11a).

Since IL-18 was the only cytokine found to have significantly higher levels in mouse serum of both syngeneic models (Fig. 5c), as well as in mouse tumors (Fig. 5d), we extended our investigation to analyze the time-interval of IMSA101-mediated IL-18 induction. PDA7940b tumor-bearing mice were treated with a single i.t. injection of IMSA101 or were left untreated for control, and tumors were harvested at different time points for IL-18 ELISA of tumor cell lysates (Supplementary Fig. 11b). By 4 h after i.t. IMSA101 injection, an increase of IL-18 levels in IMSA101-treated tumors was observed when compared to untreated tumors, reaching a significant difference at 24 h after IMSA101 treatment (Supplementary Fig. 11c). IL-18 levels continued to increase over the 6 day timecourse (Supplementary Fig. 11c, d).

To determine whether tumor macrophages contribute to higher levels of IL-18 in syngeneic mice treated with i.t. IMSA101, we performed RNA-ISH. Probes to detect the following RNAs were used: IL-18, ADGRE1 to visualize macrophages, and mouse mesothelin or human CD19 for tumor antigen detection of PDA7940b tumors or B16-huCD19 tumors, respectively (Fig. 5e). In B16-huCD19 as well as PDA7940b tumors, significantly more IL-18 RNA was detected in IMSA101-treated tumors (pooled from IMSA101-alone and IMSA101 + CART groups), when compared to control tumors (pooled from untreated and CART-alone groups) (Fig. 5f). In addition, more ADGRE1 RNA molecules were detected in IMSA101-treated tumors (Fig. 5g), while RNA-count of the tumor antigens, mouse mesothelin or human CD19, was lower in these cohorts consistent with anti-tumor response (Supplementary Fig. 11e). Notably, we observed co-expression of IL18 RNA and ADGRE1 RNA (Fig. 5h), and could detect higher levels of IL18 RNA in DAPI+ADGRE1+ cells in IMSA101-treated tumors (Fig. 5i), suggesting that IMSA101 increases macrophage count in the TIME as well as the fraction of IL-18-secreting macrophages.

Next, we performed RNA-ISH to determine whether in these models DCs also contribute to increased IL-18 levels within the TIME and used probes detecting ITGAX RNA to visualize DCs, as well as probes detecting IL-18 RNA. However, we did not observe statistical differences in the level of co-expression of IL18 and ITGAX RNA between IMSA101-treated and untreated tumors (Supplementary Fig. 11f).

Taken together, mice receiving treatment with either IMSA101 alone or in combination with CART showed higher levels of IL-18 in mouse serum in both syngeneic mouse models. Additionally, increased levels of IL-18 RNA were detected in tumors of IMSA101-treated mice, supporting that i.t. injection of IMSA101 induces IL-18 secretion.

## Increased levels of IL-18 in the TIME enhances CART function

To determine whether IL-18 signaling contributed to the increased efficacy of CART in the combination therapy group, we compared the performance of wild type CART (WT-CART) and IL-18 receptor negative CART (IL18R-neg. CART) by transducing T cells isolated from the spleen of IL-18R deficient mice (Fig. 6a). Untreated mice, as well as mice treated with IMSA101 alone were used as control.

CAR expression as well as the absence of IL-18 receptor on IL18R-neg. CART was confirmed by cell surface staining and flow cytometry (Supplementary Fig. 12a, b). No differences in T cell expansion and T cell size during manufacturing of IL18R-neg. CART and WT-CART were observed (Supplementary Fig. 12c). Prior to use in animal models, in vitro functionality of engineered cells was tested to confirm equal levels of cytokine secretion and cytotoxicity following co-incubation of CART with B16-huCD19 tumor cells. No differences in TNF, IFNγ and IL-2 levels (Fig. 6b) as well as cytotoxicity at different effector to target ratios (Fig. 6c) were seen between IL18R-neg. CART and WT-CART.

When used in syngeneic animals, significantly improved overall survival was observed in mice treated with IMSA101 + WT-CART compared to mice receiving IMSA101 + IL18R-neg. CART (Fig. 6d). However, the overall survival in the IMSA101 + IL18R-neg. CART groups was higher than mice receiving IMSA101 alone (Fig. 6d), suggesting that other factors in addition to increased levels of IL-18 in the TIME promote the IMSA101-mediated increase in CART efficacy. All twelve mice receiving IMSA101 + WT-CART cells showed complete remission of the tumor, while seven out of twelve mice cleared tumors when IMSA101 was combined with IL18R-neg. CART (Fig. 6e). In the IMSA101-alone group, three out of twelve mice showed complete remission (Fig. 6e). None of the mice receiving IMSA101 + WT-CART showed clinical signs of distress, or severe weight loss over time requiring sacrifice (Supplementary Fig. 12d). Weight loss over time was observed in some untreated mice, as well as in some mice receiving IMSA101 alone or IMSA101 + IL18R-neg. CART which correlated with tumor progression (Supplementary Fig. 12d).

In conclusion, the use of IL18R-neg. CART significantly impaired but did not abrogate anti-tumor response in mice receiving combination treatment, supporting that IMSA101 enhances CART response through IL18-mediated as well as independent mechanisms.

## Discussion

Solid tumors account for more than 90% of adult cancers[22], however CART therapies directed against solid tumor antigens have achieved limited therapeutic success due in part to the immunosuppressive nature of the TIME. One strategy to address this challenge is to combine agents that inflame the TIME so that CART can better infiltrate and function. Here, we investigate the therapeutic efficacy of combining CART with a newly developed STING agonist, IMSA101 which exhibits superior stability in human serum and increased anti-tumor reponse compared to cGAMP. These studies demonstrate that IMSA101

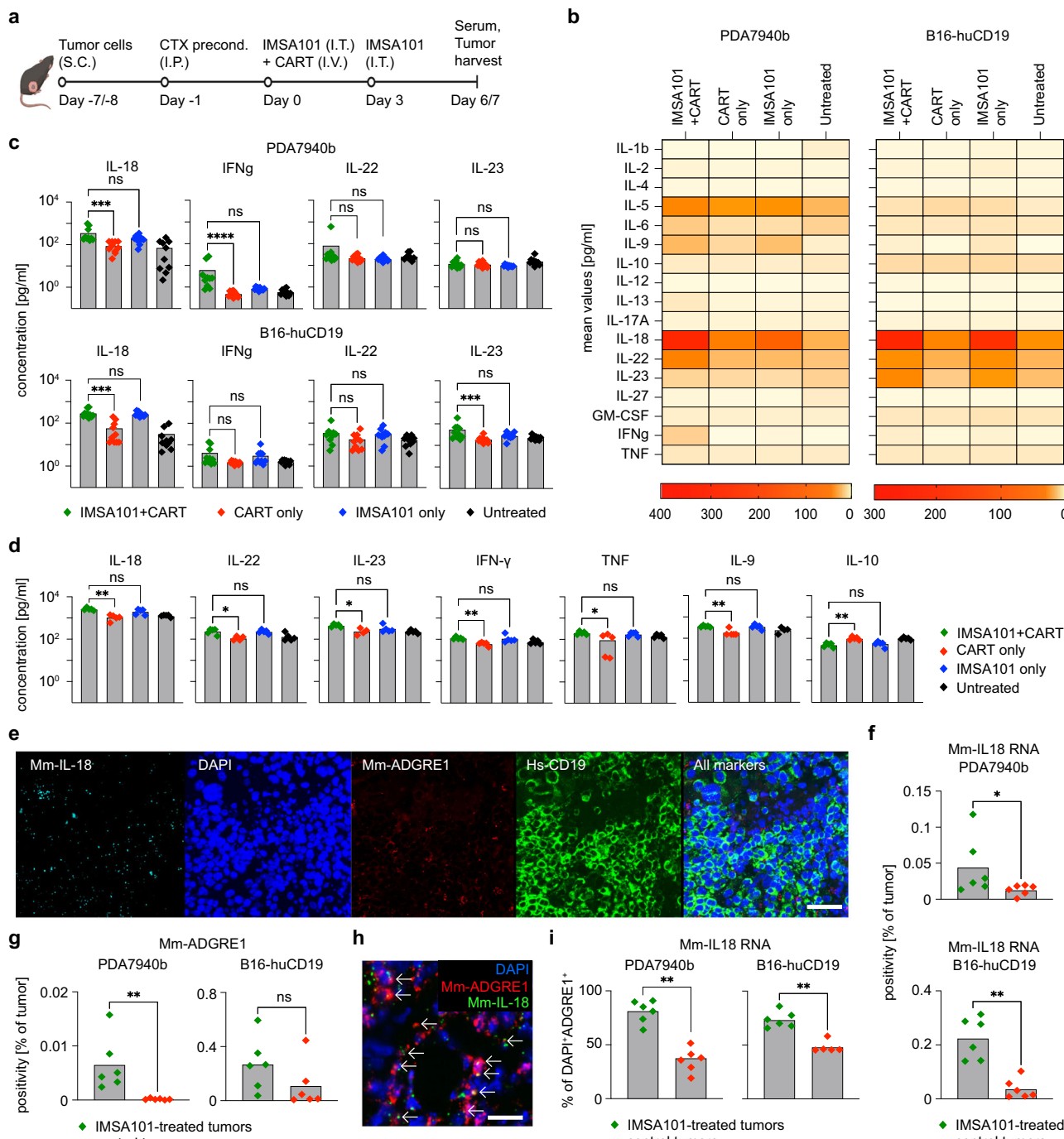

**Fig. 5 | IMSA101 induces IL-18 secretion. a** Schematic of experimental design to analyze changes in cytokine levels in mouse serum and tumors following combinatorial treatment. RNA in situ Hybridization (RNA-ISH) was performed from tumors at day 6 after treatment, tumor cytokine analyses from day 6 after treatment, and serum cytokine analyses from day 7 after treatment in separate experiments. Icons of this figure created with BioRender.com. **b** Heatmap summarizing murine cytokine levels in mouse serum (*n* = 10/cohort). Top boxes show results of each cohort as mean values. Bottom boxes show the grading scale. **c** Cytokine concentration in mouse serum as indicated (*n* = 10/cohort). Average and individual values are shown. Kruskal-Wallis one-way analysis of variance was used for statistical analysis. ns (nonsignificant) *P* > 0.05; *\*P* ≤ 0.05; \*\**P* ≤ 0.01; \*\*\**P* ≤ 0.001; \*\*\*\**P* ≤ 0.0001. **d** Cytokine concentration in mouse tumors as indicated (*n* = 5/cohort). Average and individual values are shown. Kruskal-Wallis one-way analysis

of variance was used for statistical analysis. ns (nonsignificant) *P* > 0.05; *\*P* ≤ 0.05; \*\**P* ≤ 0.01. **e** Representative photomicrographs at 20x magnification from RNA-ISH of B16-huCD19 tumor as indicated. Scale bar equal to 50 μm. **f, g** Bar graphs summerizing detectable (**f**) Mm-IL-18 RNA and (**g**) Mm-ADGRE1 RNA in tumor samples (*n* = 6/cohort). Average and individual values are shown. Two-sided Mann-Whitney U Test was used for statistical analysis. *\*P* ≤ 0.05; \*\**P* ≤ 0.01. **h** Representative photomicrograph at 40x magnification from RNA-ISH of B16-huCD19 tumor. Arrows indicate cells with co-expression of indicated markers. Scale bar equal to 20 μm. **i** Rate of detectable Mm-IL-18 RNA in DAPI⁺ADGRE1⁺ cells (*n* = 6/cohort; for B16-huCD19 control tumors group *n* = 5). Average and individual values are shown. Two-sided Mann-Whitney U Test was used for statistical analysis. \*\**P* ≤ 0.01. Source data and exact *p*-values are provided as a Source Data file.

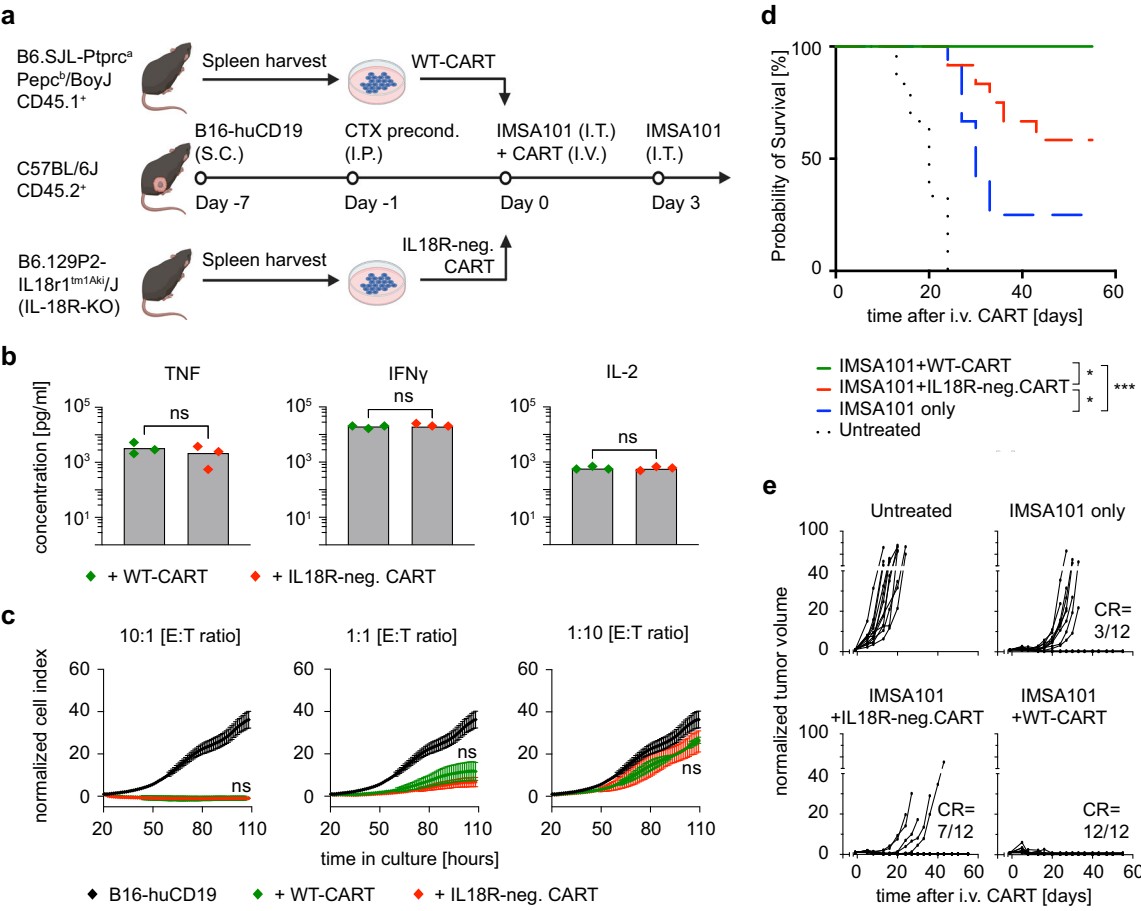

**Fig. 6 | IMSA101-induced IL-18 secretion enhances CART function and anti-tumor response in vivo. a** Schematic of experimental design for testing IL-18 receptor negative CART in a syngeneic mouse model. Icons of this figure created with BioRender.com. **b, c** In vitro assays confirm that murine CART, either wild type (WT-CART) or IL-18 receptor negative (IL18R-neg. CART) (**b**) secrete IL-2, TNF, and IFNg and (**c**) show antigen-specific cytolytic activity after co-incubation with B16-huCD19 tumor cells. Mean values ± SD of three biologically independent samples are shown. The values of the cytotoxicity assay were normalized to the time point of CART addition (approximately 24 h after seeding tumor cells). E:T ratio=effector cell to target cell ratio. Two-sided Mann-Whitney U test and ordinary two-way ANOVA was used for statistical analysis of cytokine and cytotoxicity assays, respectively. ns (nonsignificant) *P* > 0.05. **d** Kaplan-Meier survival curve (*n* = 12/cohort). Statistical significance was calculated using the log-rank Mantel-Cox test. *\*P* ≤ 0.05; \*\*\**P* ≤ 0.001. **e** Changes in tumor volume over time. CR complete remission. Source data and exact *p*-values are provided as a Source Data file.

enhances the therapeutic efficacy of CART directed against solid tumors through modulation of the TIME. Intratumoral administration of IMSA101 results in increased infiltration of CART and endogenous immune cells such as macrophages, neutrophils, and NK cells into the TIME. Further, IMSA101 treatment induces IL-18 secretion which contributes to improved CART anti-tumor function. Recent publications identified a role for STING agonists in priming and activation of inflammasomes[23,24], resulting in secretion of inflammasome-dependent cytokines, such as pro-inflammatory IL-18[25,26], in addition to cGAMP's well known function as an endogenous second messenger to induce type I interferon genes in the cytosolic STING pathway. In accordance with this, we demonstrate that IMSA101 - a small molecule acting as a cGAMP analogue - triggers IL-18 secretion.

IL-18 is an inflammasome-dependant pro-infammatory cytokine that requires caspase-1 for processing, and is produced by various cells, including macrophages and dendritic cells[27,28]. In addition to inducing the production of IFNγ, IL-18 is known to increase cytotoxicity and FAS ligand expression[29], thus playing an important role in anti-tumor immunity. While IL-18 primarily signals through its receptor IL-18R, some IL-18R-independent mechanisms were recently described in ref. 30, which could in part account for our observation that IL18R-neg. CART significantly impaired but did not abrogate anti-tumor response in mice receiving combination treatment. Further, IL-18 is known to

regulates both Th1 and Th2 responses[31], and to activate other immune cells within the TIME, such as macrophages, DCs and NK cells[28] all of which have important functions in antitumor defense.

The anti-tumor effects of IL-18 through activation of lymphocytes has been studied for many years[29], and prompted the opening of clinical trials with systemic administration of recombinant human IL-18 to treat advanced cancers, e.g., renal cell carcinoma, melanoma, or Hodgkin's lymphoma[32,33]. Although shown feasible and safe[32,33], clinical development of systemic IL-18 was discontinued due to lack of therapeutic response as a single agent[34]. Follow on investigations revealed high levels of IL-18 binding protein (IL-18BP), which negatively regulates IL-18, as the main cause for this[21,35]. In our studies, although IL-18 levels were higher in serum and tumor of IMSA101-treated mice, we did not observe differences in IL-18BP levels between cohorts receiving IMSA101 and CART, either alone or in combination.

Effects of IL-18 specifically on CART function has been investigated at our center. Importantly, we have shown that IL-18 significantly enhances the engraftment of human CD8[+] effector T cells in mice[36]. Further, we developed IL-18 secreting CART and demonstrate enhanced anti-tumor function in preclinical models[37]. This investigation product is currently being tested in a first-in-human phase I clinical trial at our center in patients with non-Hodgkin lymphoma (NHL) who had previously received and failed commercially available CART

therapy (NCT04684563)[38]. Seven patients have been treated to date, and early results show that this product is safe and did not result in new or increased side effects compared to already available CART therapies. Importantly, all patients responded to the therapy, while four of those seven patients showed complete response, suggesting that CART secreting IL-18 exhibit enhanced anti-tumor efficacy[38].

Interestingly, we also found increased IL-9 levels in IMSA101-treated tumors and upregulation of the IL-9 gene and pathway in our transcriptomics data of intratumoral CART following combination treatment. We previously showed that IL-9 signaling can help maintain T cell stemness[39]. In accordance with this, Li et al. recently reported that cGAS and STING can promote the maintenance of stem cell-like CD8$^+$ T cells, partially by regulating the transcription factor TCF1[40]. The authors treated CART with a STING agonist in vitro prior to infusion into xenograft mice and showed modest enhancement of in vivo function (36), supporting their hypothesis that overexpression of the STING pathway during the manufacturing process could be used as a possible strategy to improve CART phenotype and function. Additional upregulated pathways in intra-tumoral CART of the combination treatement group include IL-2, IL-15, and IL-21 which are known to promote T cell activation, proliferation, survival and/or CD8$^+$ T cell differentiation and maintenance[41].

Previous reports have shown that STING agonists can cause T cell death through activating cell stress and death pathways[42,43], which could represent the potential limitations of this combinatorial treatment approach. In addition, a recent publication reports that continuous induction of type I interferons can hamper CART function[44]. These studies highlight the importance of choosing the correct dosage of STING agonist, as well as the timing and order of drug administration. In our animal models, the first IMSA101 dose was given prior to and the second dose at 3 days following CART infusion which is before most CART have infiltrated the tumor. This dosing regimen enabled modulation of the TIME by IMSA101 prior to CART intra-tumoral trafficking, thereby minimizing IMSA101-induced toxicities on engineered cells. Indeed, no differences in viability of infused CART were observed between cohorts in our study.

Taken together, i.t. administration of IMSA101 improved CART trafficking into the tumor, induced intra-tumoral CART activation, and enhanced anti-tumor efficacy and overall survival in syngeneic mouse models, which was, to a significant extent, facilitated through STING agonist-mediated IL-18 induction. These studies offer compelling justification for advancing the combinatorial treatment approach into a clinic setting, which is facilitated by the fact that both products, IMSA101 (NCT04020185, NCT05846646, NCT05846659), as well as the human mesothelin-specific CAR construct used in this study (NCT03054298, NCT05623488) are currently under clinical investigation.

## Methods

All research presented in this manuscript complies with all relevant ethical regulations. The University of Pennsylvania Institutional Animal Care and Use Committee (IACUC) approved all animal experiments (protocol number: 804226), and all animal procedures were performed in the animal facility at the University of Pennsylvania in accordance with Federal and Institutional IACUC requirements.

### Study design

The purpose of this study was to test the combinatorial treatment approach using i.t. administration of the STING agonist IMSA101 and i.v. CART infusion, and to elucidate the role of IL-18 in this context. Therefore, we have first established three animal flank tumor models: Two syngeneic animal models using immunocompetent C57BL/6 mice, as well as one xenograft animal model using immunodeficient NSG mice. C57BL/6 mice allowed us to analyze the TIME as well as cell-

extrinsic mechanisms of enhanced anti-tumor CART immunity in more details, while NSG mice allowed us to investigate cell-intrinsic effects of IMSA101 on CART.

### Human tumor cells

The human PDA cell line AsPC-1 was obtained from the American Type Culture Collection (ATCC) and routinely authenticated by the University of Arizona Genetics Core. AsPC-1 cells endogenously express human mesothelin, which served as the target antigen for our CAR construct. Tumor cells were tested in regular intervals for the presence of mycoplasma contamination by the Department of Genetics at the University of Pennsylvania (MycoAlert Mycoplasma Detection Kit, Lonza). AsPC-1 cells were maintained in culture with D20 media: DMEM (1X, gibco) supplemented with 20% heat-inactivated fetal bovine serum (FBS, Seradigm), 2% 1 M HEPES buffer solution (gibco), 1% 100X glutaMAX (gibco), and 1% 10,000 U/ml penicillin + 10,000 ug/ml streptomycin (gibco).

### Murine tumor cells

The murine PDA cell line PDA7940b was established from the Kras$^{LSL.G12D/+}$p53$^{R172H/+}$ (KPC) mouse pancreatic tumor model[45] and was a kind gift from Dr. Gregory Beatty (University of Pennsylvania). PDA7940b cells endogenously express mouse mesothelin, which served as the target antigen for the CAR construct. The murine melanoma cell line B16-F10 was obtained from ATCC. B16-F10 cells were lentivirally transduced and FACS sorted to express human CD19[46], which served as the target antigen for the CAR construct used in this syngeneic animal model. The purity of human CD19 expression was routinely validated by flow cytometry. All tumor cells lines, PDA7940b, B16-F10, and B16-huCD19, were tested in regular intervals for the presence of mycoplasma contamination by the Department of Genetics at the University of Pennsylvania (MycoAlert Mycoplasma Detection Kit, Lonza). PDA7940b cells were maintained in culture with R10 media: RPMI 1640 (gibco) supplemented with 10% heat-inactivated fetal bovine serum (FBS, Seradigm), 2% 1 M HEPES buffer solution (gibco), 1% 100X glutaMAX (gibco), and 1% 10,000 U/ml penicillin + 10,000 ug/ml streptomycin (gibco). B16-F10 as well as B16-huCD19 tumor cells were maintained in culture with D10 media: DMEM (1X, gibco) supplemented with 10% heat-inactivated fetal bovine serum (FBS, Seradigm), 2% 1 M HEPES buffer solution (gibco), 1% 100X glutaMAX (gibco), and 1% 10,000 U/ml penicillin + 10,000 ug/ml streptomycin (gibco).

### Human CART

Lentiviral vectors and human CART were produced as previously described in ref. 47. For human CART production, healthy donor T cells (ND517 and ND569) were purchased from Human Immunology Core (HIC) at University of Pennsylvania. Cells purchased from this core are de-identified and are not collected specifically for this project. As a result, the University of Pennsylvania IRB has deemed that this use of human cells meets eligibility criteria for IRB review exemption authorized by 45 CFR 46.104, category #4. T cells were stimulated with anti-human CD3/CD28 antibody-coated beads (Dynabeads, gibco) at a bead:T cell ratio of 3:1. 24 h later, T cells were lentivirally transduced to express a second-generation, human mesothelin-specific CAR construct. A multiplicity of infection (MOI) of 4 was used for T cell transduction. At day six of T cell stimulation, beads were removed from culture and cells were allowed to rest down to a volume ~350 fL before cryopreservation. For cryopreservation of T cells, freezing media consisting of 90% heat-inactivated FBS (Seradigm) and 10% dimethyl sulfoxide (DMSO, SIGMA) was used. CAR expression of transduced T cells was quantified by anti-F(ab')$_2$ staining and flow cytometry one day prior to or at day of cryopreservation (for flow cytometry and antibodies see below). Human T cells were maintained in culture with R10 media (see above).

## Murine CART

Murine CART production using retroviral vectors was previously described in ref. 48. Briefly, spleen from C57BL/6 mice were harvested and T cells were purified with mouse T cell isolation beads (STEM-CELL). Purified mouse T cells were activated with anti-mouse CD3/CD28 antibody-coated beads (Dynabeads, gibco) at a bead:T cell ratio of 2:1. T cells were then retrovirally transduced to express a mouse mesothelin-specific CAR construct (MSGV-mMesoBBz) or a human CD19-specific CAR construct (MSGV-hu19BBz) on recombinant human fibronectin-coated plates (Retronectin, TaKaRa) two days after bead stimulation. Recombinant mouse IL-2 (50 U/ml) was supplemented at day 1 and then supplemented with fresh mouse T cell media ( = RPMI 1640 (gibco) supplemented with 10% heat-inactivated FBS (Seradigm), 1% 100X glutaMAX (gibco), 1% 10,000 U/ml penicillin + 10,000 ug/ml streptomycin (gibco), 1X 100 mM sodium pyruvate (gibco), and 50 uM β-mercaptoethanol (SIGMA)) containing 50 U/ml IL-2 every day. At day five of stimulation, mouse CART were harvested, de-beaded, analyzed for CAR expression by anti-F(ab')$_2$ staining and flow cytometry (for flow cytometry and antibodies see below), and used for in vivo experiments the same day.

## IMSA101

The STING agonist IMSA101, an analogue of 2'3'-cGAMP, was developed, chemically synthesized, and provided by ImmuneSensor Therapeutics. IMSA101 is a cyclic di-nucleotide derivative of cGAMP with improved serum stability and anti-tumor efficacy. US patents covering STING agonists including IMSA101 are filed (10,336,786 and 10,519,188). Detailed description of synthesis of a related cGAMP analogue, IMSA172, was previously reported[49]. IMSA101 comply with community criteria and requirements. IMSA101 is not available for research purposes by MTA, but collaboration agreements will be considered. Please contact Dr. Lijun Sun directly (jsun@immunesensor.com).

## Reporter assays for STING activation in cells

THP1-ISG-luc and Raw-ISG-luc reporter cells were purchased from Invivogen. STING-KO cells were generated using CRISPR technology[50]. These cells were cultured in RPMI-1640 with 10% FBS and antibiotics. Cellular activity of cyclic di-nucleotides were tested in reporter cells as described previously in ref. 49. IMSA101-induced IFNβ production was quantified using an ELISA kit (Invivogen).

## IMSA101 stability assay in human serum

A mixture containing 100μM of IMSA101 or cGAMP, 50% human serum (Sigma Aldrich), 10 mM NaHPO4 (pH7.2), and 100 mM NaCl was incubated at 37 C for 1, 2, 4, and 8 h, followed by heating at 95 C for 5 min. The mixture was then centrifuged at 10,000 g for 5 min. Activity in supernatant was measured in THP1-ISG-luc cells as described above.

## In vitro cytokine secretion assay from supernatants

CART (IL18R-neg. and WT) were stimulated overnight with the tumor cell lines B16-huCD19 at a 1:1 effector:target cell ratio. Concentrations of the cytokines IL-2, TNF, and IFNg in the supernatants were analyzed using the mouse Th1/Th2 cytokine CBA kit for mouse T cells (BD Biosciences) according to manufacturer's instructions. Following acquisition of sample data using Fortessa LSRII flow cytometer (BD Biosciences) equipped with the FACSDiva software (BD Biosciences), the sample results were generated using the BD CBA Analysis Software (BD Biosciences).

## In vitro cytotoxicity assay

In vitro analysis of CART killing (IL18R-neg. and WT) was performed using a real-time, impedance-based assay with xCELLigence Real-Time Cell Analyzer System (ACEA Biosciences). Briefly, 2.5e + 3 B16-huCD19 cells were seeded to a 96-well E-plate. The next day, CART were added to the wells in a 10:1, 1:1, and 1:10 effector to target cell (E:T) ratio. Tumor killing was monitored over a total of 10 days. For data analysis, cell counts were normalized to the time point of CART addition (approximately 24 h after start of experiment).

## Mice

The University of Pennsylvania Institutional Animal Care and Use Committee (IACUC) approved all animal experiments (protocol number: 804226), and all animal procedures were performed in the animal facility at the University of Pennsylvania in accordance with Federal and Institutional IACUC requirements. NSG mice were originally procured from Jackson Laboratories and bred by the Stem Cell & Xenograft Core (SCXC) at University of Pennsylvania. For syngeneic mouse experiments, C57BL/6 mice, B6.SJL-Ptprc$^a$ Pepc$^b$/BoyJ mice, and B6.129P2-IL18r1$^{tmlAki}$/J mice were obtained from Jackson Laboratories. Six- to eight-week-old female mice were used for in vivo experiments. Mice were maintained under pathogen free conditions. Schemas of used mouse models are shown in detail in each relevant main and Supplementary Fig. One day prior treatment, tumor size was measured in all in vivo experiments to calculate tumor volume. Mice were then categorized into cohorts with equal average tumor sizes. Animals with rapid or slow tumor growth were excluded from the study prior to IMSA101 and/or CART administration. Data analysis was based on objectively measurable data in an unblinded fashion. Mice were subject to routine veterinary assessment for signs of overt illness and were euthanized at experimental termination or when predetermined IACUC rodent health endpoints were reached. The maximal tumor size of 2 cm in diameter permitted by the institutional review board was not exceeded.

## Xenograft animal model

AsPC-1 cells were used for in vivo experiments with NSG mice. 2e + 6 AsPC-1 cells in a total volume of 100 ul of a 1:1 matrigel (Corning): 1X DPBS (Gibco) mix were implanted subcutaneously into the right flank of NSG mice. Tumor size was allowed to reach approximately 150–200 mm³. At day 15, IMSA101 at a dose of 3 μg diluted in 50 μl of PBS was injected i.t. into mouse tumors. Following IMSA101 administration, mice were given 3e + 5 i.v. human mesothelin-specific CART the same day. A second i.t. dose of IMSA101 (3 μg in 50 μl PBS) was given three days later. For control, mice were either treated with i.t. IMSA101 only, with i.v. CART only, or were left untreated. Caliper, as well as mouse body weight measurements were performed at least once a week. Values for tumor size and body weight were normalized to day −1 in main and Supplementary Figs. In separate experiments, NSG mice were sacrificed at day 17 after treatment for staining of CD45$^+$ cells in mouse tumors and peripheral blood and for pathology/IHC of mouse tumors (for details see below). Health monitoring followed IACUC body scoring system (BCS) guidelines.

## Syngeneic animal models

For testing the combinatorial treatment approach IMSA101 + CART, C57BL/6 mice were inoculated with 5e + 5 PDA7940b or B16-huCD19 tumor cells in a total volume of 100ul 1X DPBS (Gibco) subcutaneously into the right flank. At day six or seven in the B16-huCD19 or PDA7940b models, respectively, mice underwent lymphodepletion with i.p. cyclophosphamide (Sigma-Aldrich) at a dose of 120 mg/kg. I.t. IMSA101 administration at a dose of 3 μg in 50 μl of PBS followed by i.v. CART infusion (5e + 6 viable CART in 150 ul PBS) was given the next day when tumor size was approximately 50 mm³. Three days later, another i.t. injection of IMSA101 at the same dose was given. Mice received either i.t. IMSA101 only, i.v. CART only, i.t. IMSA101+i.v. CART, or were left untreated for control. For tumor re-challenge experiments, C57BL/6 mice were implanted with 5e + 5 B16-F10 tumor cells s.c. into the contralateral left flank, and treatment-naïve mice were used for control. For dual-flank tumor models, C57BL/6 mice were implanted with

5e + 5 and 2e + 5 B16-huCD19 cells into the right and left flanks, respectively. Only the right flank tumors were treated with i.t. IMSA101 in those experiments. For in vivo comparison of IMSA101 with cGAMP, 1e + 6 B16-F10 tumor cells were injected s.c. into the right flank of C57BL/6 mice, and s.c. injections of 3 μg and 10 μg IMSA101 and cGAMP were performed at day four, day eight, and day 14 after tumor inoculation. Caliper as well as mouse body weight measurements were performed at least twice weekly for all animal experiments. Values for tumor size and body weight were normalized to day -1 in main and Supplementary Figs. where indicated. In defined experiments, C57BL/6 mice were sacrificed at day six or seven after treatment for i) isolation of intratumoral CD45.1⁺ cells for gene expression and pathway analyses using the Nanostring nCounte platform, for ii) cell surface staining of CART and other immune cells in flow cytometry from mouse tumors, spleen, and peripheral blood, for iii) pathological assessment, IHC, and RNA in situ hybridization (RNA-ISH) of mouse tumors, as well as for iv) cytokine analysis from mouse serum and tumors (for details see below). Health monitoring followed IACUC body scoring system (BCS) guidelines.

### Caliper measurements of subcutaneous tumors

Tumor size was measured with calipers and the volumes were calculated as follows: volume = (length in mm × width$^2$ in mm)/2. Values for tumor size were normalized to day -1 in main and Supplementary Figs. where indicated. For the in vivo experiment comparing IMSA101 with cGAMP the following formula was used to calculate tumor volumes: volume = length (in mm) x Width (in mm) x Height (in mm) x 3.14/6.

### Preparation of single cell suspension from tumor and spleen

Tumors and spleen were harvested and processed into single cell suspensions as previously described in ref. 39. In short, following tumor harvest, tumor tissue was minced into 3–5 mm pieces using a scalpel and razor blade. Minced tissue was then incubated in DMEM (1X, gibco) supplemented with 1X Collagenase/Hyaluronidase (STEMCELL) and DNase I Solution (1 mg/mL, STEMCELL) at 37 °C shaking at 200 rpm for 30 min. Red blood cells were lysed using ACK Lysis Buffer (Life Technologies) prior to use in flow cytometer analysis or magnetic and/or FACS sorting of intratumoral CART for gene expression and pathway enrichment analyses. Spleen were minced by using the flat end of a syringe plunger, and red blood cells were lysed using ACK Lysis Buffer (Life Technologies) prior to flow analysis.

### Transcriptomic analyses of intratumoral CART

Tumors were harvested and single cell suspensions were prepared as described above. Intratumoral CART (murine CD45.1⁺ cells in the PDA7940b syngeneic animal model) were stained using FITC-labelled anti-mouse CD45.1 antibody (Miltenyi Biotech, CAT: 130-124-211) and magnetically sorted using the EasySep™ Mouse FITC Positive Selection Kit II according to the manufacturer's instructions (STEMCELL). RNA extraction of cells, as well as NanoString nCounter was then performed by the Genomics Facility at the Wistar Institute at University of Pennsylvania. The nCounter® Immune Exhaustion Panel (NanoString) was used for this experiment which includes 785 genes across 47 pathways involved in immune activation, immune suppression, immune status, immune checkpoint, epigenetics, and metabolism and microenvironment. The nCounter results were processed and normalized using the nSolver Analysis software according to manufacturer protocol (for normalized counts of the full NanoString nCounter dataset see Supplementary Data 3). Log2 normalized counts were used to calculate Log Fold Change between groups and graphed using the pheatmap and ggplot2 Bioconductor R packages. Gene lists were retrieved from Reactome pathways (https://reactome.org). Gene Set Variance Analysis (GSVA) was performed using the GSVA Bioconductor R package and coverage thresholds were set to 50 percent or greater[51]. GSVA results

were graphed using the ggplot2 Bioconductor R package. For statistical analysis, two-sided t-test with equal variances was used.

### Peripheral blood stain for (CAR) T cells

Peripheral blood of C57BL/6 mice or NSG mice was obtained by cardiac puncture, stained, and cell numbers of cells were quantified using TruCount tubes (BD Biosciences) according to the manufacturer's instructions. Following acquisition of sample data using Fortessa LSR II flow cytometer (BD Biosciences) equipped with the FACSDiva software (BD Biosciences), sample results were analyzed using FlowJo version 10 software (BD Biosciences).

### Cytokine analysis from mouse serum and mouse tumors

Mouse serum and supernatant of tumor tissue lysates of C57BL/7 mice were submitted to the Human Immunology Core (HIC) at University of Pennsylvania for Luminex assay. The Th1/Th2/Th9/Th17/Th22/Treg Cytokine 17-Plex Mouse ProcartaPlex™ Panel (Invitrogen) was used for detection of mouse cytokines. To obtain mouse serum, whole blood was collected via cardiac puncture, left at room temperature for approximately 20 min, and centrifuged to remove clots. For preparation of tumor cell lysates, approximately 0.5 g of tumor tissue was homogenized on ice in 0.6 ml RIPA buffer (Thermo Scientific) supplemented with 1X protease and phosphatase inhibitor cocktail (Thermo Scientific). Supernatants of tumor cell lysates were obtained following centrifugation of homogenates. For analysis of IL-18BP from mouse serum and IL-18 from tumor cell lysates at different time points, a Mouse IL-18BP ELISA Kit and a Mouse IL-18 ELISA kit (abcam), respectively, was used according to manufacturer's instructions and measured in Synergy H4 hybrid multimode microplate reader (BioTek).

### Pathological assessment and immunohistochemistry (IHC)

Tumors of NSG or C57BL/6 mice were harvested and prepared for standard pathological and/or immunohistochemistry (IHC) analysis by the Comparative Pathology Core (CPC) at the University of Pennsylvania School of Veterinary Medicine. Formalin fixed tissues were routinely processed for paraffin embedding, sectioning, and staining for hematoxylin and eosin (H&E). For immunohistochemistry, 5 μm thick paraffin sections were mounted on ProbeOn™ slides (Thermo Fisher Scientific). The immunostaining procedure was performed using a Leica BOND RXm automated platform combined with the Bond Polymer Refine Detection kit (Leica). Briefly, after dewaxing and rehydration, sections were pretreated with the epitope retrieval BOND ER2 high pH buffer (Leica) for 20 min at 98 °C. Endogenous peroxidase was inactivated with 3% $H_2O_2$ for 10 min at room temperature (RT). Non-specific tissue-antibody interactions were blocked with Leica Power-Vision IHC/ISH Super Blocking solution (PV6122) for 30 min at RT. The same blocking solution also served as diluent for the primary antibodies. Rabbit monoclonal primary antibodies against F4/80 (CST, CAT: 70076), FoxP3 (CST, CAT: 12653), Klrb1c/CD161c (CST, CAT: 39197), a rabbit polyclonal antibody against Ly-6G (CST, CAT: 87048), and a rat monoclonal antibody against CD3ε (Bio-Rad, CAT: MCA1477T), were used at a concentration of 1/1000, 1/300, 1/500 and 1/600, respectively. Antibodies were incubated on the sections for 45 min at RT. A biotin-free polymeric IHC detection system consisting of HRP-conjugated anti-rabbit or anti-rat IgG was then applied for 25 min at RT. Immunoreactivity was revealed with the diaminobenzidine (DAB) chromogen reaction. Slides were finally counterstained in hematoxylin, dehydrated in an ethanol series, cleared in xylene, and permanently mounted with a resinous mounting medium (Thermo Scientific ClearVueTM coverslipper). Sections of human tonsil and pooled mouse lymphoid tissues were included as positive controls. Negative controls were obtained by replacing the primary antibody with an irrelevant isotype-matched antibody. The slides were scanned using the Aperio Versa 200 whole slide scanner (Leica Biosystems,

Buffalo Grove, Illinois), and the image analysis was performed using the ImageScope software (Leica Biosystems, Buffalo Grove, Illinois). Nuclear algorithms for cell counting were generated for the CD3, FoxP3 and Klrb1c/CD161c stains, and positive pixel count algorithms for area quantification were generated for the Ly-6G and F4/80 stains since staining properties of these two markers made the quantification of individual cells inaccurate and unreliable. Tissue sections were analyzed by a board-certified veterinary pathologist in a blinded fashion, and details concerning experimental design and tested compounds were revealed only at the end of study. The severity of tumor necrosis, intra- and peri-tumoral inflammation, and intra-tumoral fibrosis were scored in a semi-quantitative way by the pathologist and results were presented in the form of a heat-map. The scoring of each parameter was performed as follows: i) no, ii) minimal, iii) mild, iv) moderate, and v) severe change. Photomicrographs at a 2x or 4x magnification were acquired using the Olympus DP22 digital camera with the Olympus cellSens digital imaging software. Over 10 sections were obtained from the paraffin blocks of the tumors for various histochemical and immunohistochemical stains, which makes some tumor sections appear differently across the different sections (Fig. 2g).

### RNA in situ hybridization (RNA-ISH)
RNA-ISH to visualize RNA molecules in individual cells of formalin-fixed, paraffin-embedded tumor tissues was performed according to manufacturer's instructions (RNAscope, ACDBio). The following RNAscope probes were used: Hs-CD19 (CAT: 402711, ACDBio), Mm-Adgre1-C2 (CAT: 460651-C2, ACDBio), Mm-IL18-C3 (CAT: 416731-C3, ACDBio), Mm-Msln (CAT: 443241, ACDBio), Mm-Itgax-C2 (CAT: 311501-C2, ACDBio), vMSGV-C3 (CAT:1140971-C3, ACDBio), 4-plex Negative Control Probe (CAT: 321831, ACDBio). The 4-plex Negative Control Probe was used to confirm that the samples were free of non-specific signal or interfering substances that would confound interpretation and quantification. The RNAscope slides were scanned using the immunofluorescence filters on the same Aperio Versa scanner, and positive pixel count algorithms for area quantification were generated for each probe using the ImageScope software. Using the same ImageScope software, a cell counting algorithm for immunofluorescence was generated and used to count the number of macrophages/DCs (DAPI + Adgre1/Itgax RNA), as well as cells producing IL-18 (DAPI + IL18 RNA) and macrophages/DCs producing IL-18 (DAPI + Adgre1/Itgax RNA + IL18 RNA). The number of cells were normalized to the area of analysis. Cell counting algorhythms were generated and alayses were performed by a board-certified veterinary pathologist in a blinded fashion, and details concerning experimental design and tested compounds were revealed only at the end of study. Photomicrographs at a 20x or 40x magnification were acquired using the Olympus DP22 digital camera with the Olympus cellSens digital imaging software.

### Flow cytometer and antibodies
All markers were stained in 1X DPBS (gibco) containing 5% heat-inactivated FBS (Seradigm). Human mesothelin expression on the human tumor cells AsPC-1 was detected with biotin anti-human mesothelin (1/25 dilution; clone MB; CAT: 530203; BioLegend). Biotin mouse IgG2a, κ isotype ctrl antibody (1/25 dilution; clone MOPC-173; CAT: 400203; BioLegend) was used for isotype control. Human CD19 expression on B16-huCD19 cells was analyzed using PE/Dazzle 594 anti-human CD19 antibody (1/100 dilution; clone HIB19; CAT: 302252; Biolegend). Monoclonal rat anti-mouse MSLN / Mesothelin antibody (1/25 dilution; clone: B35; CAT: LS-C179484; LSBio) and PE mouse anti-rat IgG2a antibody (1/100 dilution; clone: r2a-21B2; CAT: 12-4817-82; Invitrogen) were used to stain for mouse mesothelin expression of the murine tumor cells PDA7940b. For detection of human T cells from peripheral blood and tumor single cell suspensions

of NSG mice, Brilliant Violet 605 anti-human CD45 antibody was used (1/80 dilution; clone 2D1; CAT: 368524; BioLegend). Mouse CART, endogenous T cells, B cells, NK cells, M1/M2-macrophages, MDSCs, and $T_{regs}$ were detected in peripheral blood, spleen, or tumors of syngeneic mice by using following antibodies: Alexa Fluor 700 anti-mouse CD45 (1/200 dilution; clone 30-F11; CAT: 103128; BioLegend), Brilliant Violet 650 anti-mouse CD3 (1/40 dilution; clone 17A2; CAT: 100229; BioLegend), Brilliant Violet 510 anti-mouse CD45.1 (1/50 dilution; clone A20; CAT: 110741; BioLegend), Brilliant Violet 785 anti-mouse CD19 (1/50 dilution; clone 6D5; CAT: 115543; BioLegend), PE/Dazzle 594 anti-mouse NK1.1 (1/100 dilution; clone PK136; CAT: 108748; BioLegend), Alexa Fluor 647 anti-mouse CD86 (1/100 dilution; clone GL-1; CAT: 105020; BioLegend), PerCP/Cyanine 5.5 anti-mouse F4/80 (1/50 dilution; clone BM8; CAT: 123128; BioLegend), Brilliant Violet 510 anti-mouse I-A/I-E (1/100 dilution; clone M5/114.15.2; CAT: 107636; BioLegend), Brilliant Violet 711 anti-mouse/human CD11b (1/100 dilution; clone M1/70; CAT: 101242,;BioLegend), Brilliant Violet 421 anti-mouse Ly-6C (1/50 dilution; clone HK1.4; CAT: 128032; BioLegend), Alexa Fluor 488 anti-mouse CD25 (1/100 dilution; clone PC61; CAT: 102017; BioLegend), Brilliant Violet 421 anti-mouse FoxP3 (1/100 dilution; clone MF-14; CAT: 126419; BioLegend), and Brilliant Violet 421 Rat IgG2b Isotype Ctrl (1/100 dilution; clone RTK4530; CAT: 400639; Biolegend). For intracellar staining of FoxP3 and Isotype Ctrl, Cyto-Fast Fix/Perm Buffer Set was used (CAT: 426803; BioLegend). Mouse CD4, CD8, PD-1, CD44, and CD62L were deteted using the follwoing antibodies: Brilliant Violet 711 anti-mouse CD4 (1/200 dilution; clone RM4-5; CAT: 100550; BioLegend), Brilliant Violet 605 anti-mouse CD8a (1/160 dilution; clone 53-6.7; CAT: 100744; BioLegend), Brilliant Violet anti-mouse CD279/PD-1 (1/50 dilution; clone 29 F.1A12; CAT: 135218; BioLegend), PE/Cyanine7 anti-mouse/anti-human CD44 (1/100 dilution; clone IM7; CAT: 103030; BioLegend), and FITC anti-mouse CD62L (1/100 dilution; clone MEL-14; CAT: 104406; BioLegend). TruStain FcX anti-mouse CD16/32 (CAT: 101320; BioLegend) or Human TruStain FcX Fc Receptor Blocking Solution (CAT: 422302; BioLegend) was used prior staining of single cell suspensions of mouse or human tumors and spleen, respectively. Cells were stained for viability using Invitrogen™ LIVE/DEAD™ Fixable Near-IR Dead Cell Stain Kit (CAT: L10119; Invitrogen) according to manufacturer's instructions. Biotin-SP (long spacer) AffiniPure F(ab')$_2$ fragment goat anti-human IgG (1/25 dilution; CAT: 109-066-006; Jackson ImmunoResearch) was used to detect CAR constructs on human and murine T cells. APC anti-mouse CD218a (IL-18Rα) antibody was used to stain for IL-18R on mouse T cells (1/100 dilution; clone A17071D; CAT: 157906; BioLegend). For all biotinylated antibodies, PE Streptavidin (1/100 dilution; CAT: 554061; BD Biosciences) was used prior to flow cytometer analysis. Where indicated, CountBright Absolute Counting Beads (CAT: C36950; Invitrogen) were used to get absolute cell counts as per manufacturer's instructions. All data were collected by a LSRFortessa cytometer (BD Biosciences) equipped with the FACSDiva software (BD Biosciences) and analyzed using FlowJo version 10 software (BD Biosciences).

### Statistical analysis
Statistical analysis was performed with Prism version 9 (GraphPad Software). Each figure legend denotes the statistical test used. All central tendencies indicate the mean, and all error bars indicate standard deviation (SD). Survival curves were drawn using the Kaplan-Meier method and the differences of 2 curves were compared with the log-rank Mantel-Cox test. Mann-Whitney U Test was used to compare 2 groups and Kruskal-Wallis one-way analysis of variance was used to compare 3 or more groups. Two-way ANOVA with Sidak's test was used for pairwise multiple comparisions. For all figures, ns indicates $p > 0.05$ (non-significant), * indicates $p \le 0.05$, ** indicates $p \le 0.01$, *** indicates $p \le 0.001$, and **** indicates $p \le 0.0001$. Graphs were created by Prism version 9 (GraphPad Software), Adobe Illustrator (Adobe), and created with BioRender.com.

**Reporting summary**

Further information on research design is available in the Nature Portfolio Reporting Summary linked to this article.

## Data availability

The authors declare that all data of this study are available within the article, Supplementary Information or Source Data file. Supplementary Data are also available in Figshare [https://doi.org/10.6084/m9.figshare.24183867]. Source data are provided with this paper.

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

## Acknowledgements

We thank Tong Da, Carolyn Shaw, Yujie Ma, John Scholler, Enrico Radaelli, and Yuanwei Dai for technical assistance and/or helpful discussions. We also thank the Human Immunology Core (HIC) at University of Pennsylvania for reliable supply of healthy human T cells and for performing the Luminex cytokine assays, the Genomics Facility at the Wistar Institute for RNA extraction and performing NanoString nCounter assays, and the Comparative Pathology Core (CPC) at School of Veterinary Medicine for pathological assessment, immunohistochemistry, image acquisition, and analysis. The Penn Vet Comparative Pathology Core is supported by the Abramson Cancer Center Support Grant (P30CA016520). The scanner used for whole slide imaging and the image acquisition software was supported by an NIH Shared Instrumentation Grant (S10OD023465-01A1). The Leica BOND RXm instrument used for IHC was acquired through the Penn Vet IIZD Core pilot grant opportunity 2022. UU was supported by a Mildred-Scheel-Postdoctoral Fellowship of the German Cancer Aid. RMY and CHJ were supported by NIH grant P01CA214278.

## Author contributions

Conceptualization: Z.J.C., C.H.J., Methodology: U.U., L.S., R.M.Y., Investigation: U.U., S.C., A.V.F., C.A.A., Visualization: U.U., Funding acquisition: U.U., R.M.Y., C.H.J., Project administration: U.U., R.M.Y., Supervision: R.M.Y., Z.J.C., C.H.J., Writing – original draft: U.U. Writing – review & editing: S.C., A.V.F., C.A.A., L.S., R.M.Y., Z.J.C., C.H.J., Sofia Castelli and Amanda V. Finck contributed equally.

## Competing interests

R.M.Y. and C.H.J. are inventors on patents and/or patent applications licensed to Novartis Institutes of Biomedical Research and receive license revenue from such licenses. C.H.J. is an inventor on patents and/or patent applications licensed to Kite Pharma, Capstan Therapeutics. Dispatch Therapeutics and BlueWhale Bio. C.H.J. is a member of the scientific advisory boards of AC Immune, BluesphereBio, BlueWhale Bio, Cabaletta, Carisma, Cartography, Cellares, Celldex, Decheng, Poseida, Replay Bio, Verismo, and WIRB-Copernicus. L.S and Z.J.C. are inventors of US patents covering STING agonists including IMSA101 (10,336,786 and 10,519,188). L.S. is an employee of ImmuneSensor Therapeutics. Z.J.C. is a scientific collaborator with Pfizer and ImmuneSensor Therapeutics, and a scientific advisor for Brii Biosciences. No competing interests were declared by the other authors.
