## [Peer Review File · Nature Communications]

The STING agonist IMSA101 enhances chimeric antigen receptor T cell function by inducing IL-18 secretionREVIEWER COMMENTS

Reviewer #1 (Remarks to the Author): with expertise in CAR-T

In this manuscript, Uslu et al. enhance the efficacy of CAR-T therapy by STING agonist. Interestingly, authors found that IL-18 is contributed to the enhancement. However, some issues should be addressed before publishing this work.

1. Previous investigations have reported that the combination of CAR-T cells with STING agonists can boost the efficacy of this immunotherapy (.doi: 10.1084/jem.20200844.; DOI:10.1136/jitc-2021-003351) The novelty of this work is hampered.
2. Macrophages were attributed to increase the level of IL-18. Many cells can produce IL-18 under specific circumstances. Current data only support macrophage secretion is one of the sources of IL-18. Authors should draw a safe conclusion of such description.
3. IL-18 is a complex cytokine. IL-18Ra knock-out mice have been widely used to investigation the function of IL-18. However, recently published work showed that other IL-18R independent mechanisms can elucidate the function of IL-18. In current version of manuscript, authors should pay attention to the IL-18 independent pathways to comprehensively interpret the findings in this manuscript.
4. Detailed description of IMSA101 is lacked. It is strongly suggested that authors add more info. about this novel sting agonist. It is easy to cite more references including the preclinical data of IMSA101 to the potential readers.

Reviewer #2 (Remarks to the Author): with expertise in STING, cancer immunology

CART cell therapy demonstrated clinical benefit in the treatment of hematological tumors. However, clinical investigations revealed that this therapeutic venue was ineffective against solid tumors. The present study aims to address this issue and proposes a combination strategy to enhance the efficacy of CART cell therapy against solid tumors. The strengths of the present study rely on the potential clinical relevance and on the in vivo results obtained. The results clearly indicate that in the presented settings the combination of a novel STING agonist (IMSA101) and CART cell treatment yields potent anticancer effects. There are unfortunately also several limitations that detract from the quality of the work. The main weakness of the work is the absence of a conceptual advance. The work of Conde et al.

(JITC, 2021) showed that the combination of a STING agonist with CART cell therapy triggered anticancer effects. Specific comments follow.

1) The authors show data indicating that IMSA101 has a superior stability compared to cGAMP. While this is certainly of interest, how ISMA101 specifically affects STING signaling is not reported here. As mentioned by the authors, the strength of STING signaling will differentially affect all immune cell subsets both directly and indirectly. What is the effect of ISMA101 on T cells for instance? Will it drive cell T cell death? Will it enhance T cell activation? Will it favor T cell differentiation into effector or regulatory subsets? These questions have been addressed with cGAMP (please see, for instance, in addition to the work of Conde et al., JITC, 2021, Kuhl et al., EMBO Rep., 2023 and Ni et al., JITC, 2022). In the absence of additional information on IMSA101, and in vivo comparisons using cGAMP, it is difficult to assess the potential clinical translation of the presented combination.

2) The results are compelling and supportive but considering the previous work of Conde et al. (JITC, 2021), it would have been of interest to document why IL-18 is especially important in that context. Is it due to IMSA101 or to the specific settings of the presented experiments?

3) The authors present insufficient evidence to claim that macrophages are among the major cell types that secrete IL-18 in response to the combination therapy. What is their status? Are DCs involved as well? Is an activation of the inflammasome involved? Additional molecular studies are required.

4) Are T cell functions intrinsically affected by the deficiency of IL-18 signaling? As fairly acknowledged by the authors, the contribution of IL-18 is partial in the presented experiments and additional explorations are required.

In summary, the authors have demonstrated in two preclinical models that the combination IMSA101 and CART cell therapy is effective against solid tumors. While these results are interesting, the conceptual advance here presented is limited regarding the large number of studies that have previously addressed the relevance of combination therapies using STING ligands.

Reviewer #3 (Remarks to the Author): with expertise in CAR-T, cancer immunology

Study key findings:

The authors used tumor bearing syngeneic models to show combination i.t. STING agonist (IMSA101) and CART cells was significantly more effective at controlling tumour growth and improved mouse survival, than CART or IMSA101 alone. The responding mice were resistant to re-challenge and had evidence of epitope spreading. Increased tumor-infiltrating endogenous immune subsets and CAR-T cells were present in the PDA7940b (but not B16-huCD19) tumors when co-treated with IMSA101. Combination CART + i.t. IMSA101 treatment showed significantly increased expression of genes in T cell activation and cytokine signalling (IL18 pathway highest). Mouse serum IL18 was significantly higher in the cohort treated with CART + i.t. IMSA101 in both models, transcript for IL18 and ADGRE1 (CD68) were co-located. IL18R KO CAR-T cells were less effective than WT CAR-T cells at controlling B16-huCD19 tumour growth, when used in combination with i.t. IMSAA101.

Comments:

Figure 2G. The IHC images show a differential distribution of LY6G+ and F4/80+ cells in treated tumors. Are these images from serial sections of the same tumors? Are all combination treated tumors characterized by excluded F4/80+ cells?

Figure 3 shows nanostring GEP data for [IMSA101 + CART] vs CART alone. This is very interesting and shows the combination treatment induces increased expression of genes for T cell immune checkpoint molecules, pro-inflammatory chemokines, IFNA1 and gzmB. However IFNg does not appear to be differentially expressed between these two treatment cohorts, and suggests the IFN-g response pathway is not critical for the increased anti-tumor effect. Can the authors provide data which explains the trafficking signals necessary for the CAR-T cell and endogenous immune cell trafficking into these tumors?

Figure 4 shows combination treatment induced IL18 production in macrophages in the tumor. Can the authors provide quantitative data to show the % and/or number of macrophages that make IL18 compared to untreated or monotherapy tumors? What is the time interval for the increased IL18 production?

Also the combination drug-treated mice have increased IL18 in the serum (B-C). Can the authors also provide data exploring the same cytokines in the tumor?

Figure 5 indicates that IL18 response by CAR-T cells plays a role in the anti-tumor efficacy of combination i.t. IMSA101 + CAR-T therapy. There are a series of questions that emanate from this finding:-

1. What effect does IL-18 have on the functional polarisation of the tumour infiltrating CAR-T cells? Is there a difference in CD4+ vs CD8+ CAR-T cell response to IL18?
2. Which endogenous immune cell types in the tumor express IL18R? What effect does IL18 signalling have on their functional responses in the tumour?
3. A number of cytokine signalling pathways are induced by IMSA101 + CAR-T therapy in the tumor (Fig 5). These may also play a critical role in the anti-tumour efficacy of this combination treatment. This could either be in concert with IL18 or as a parallel driver of efficacy. Can the authors provide data to show which of the upregulated cytokine pathways are critical for the anti-tumour effect?
4. What time interval is critical for IMSA101-induced IL-18 production in the TME to induce the anti-tumor effect?

Minor:

1. For the in vivo studies, please indicate in the legends whether the data is representative or pooled and from how many experiments.
2. Fig 2F, can the authors please explain why the units for F4/80+ and LY6G+ cells are pixels/mm² rather than cells/ mm².

Reviewer 1:

In this manuscript, Uslu et al. enhance the efficacy of CAR-T therapy by STING agonist. Interestingly, authors found that IL-18 is contributed to the enhancement. However, some issues should be addressed before publishing this work.

Response:

We highly appreciate the reviewers' comments and suggestions and wish to thank the reviewer for the time and effort. We have followed the reviewer's suggestions and changed the manuscript as outlined below.

Comment 1:

Previous investigations have reported that the combination of CAR-T cells with STING agonists can boost the efficacy of this immunotherapy (.doi: 10.1084/jem.20200844.; DOI:10.1136/jitc-2021-003351). The novelty of this work is hampered.

Response to comment 1:

We thank the reviewer for this comment and agree that there are prior publications on CART and STING agonists (see references 14,15). In this study, we combined CART with a newly developed and previously unpublished STING agonist, IMSA101 which is a cGAMP analogue and which shows enhanced serum stability and superior *in vivo* anti-tumor efficacy (shown now in the manuscript) compared to conventional cGAMP. Therefore, combining IMSA101 with CART represents an attractive and highly translational approach. Further, we have analyzed IMSA101-mediated changes of the TIME; in particular cytokine levels in serum and tumors as well as transcriptomic changes of intratumoral CART including pathway analyses, which has not been reported in previous publications. We found that enhanced CART anti-tumor function was in part enhanced through IMSA101-mediated IL-18 induction and thus, our study provides new mechanistical insights into how STING agonists improve CART function and extends findings of previous publications.

Comment 2:

Macrophages were attributed to increase the level of IL-18. Many cells can produce IL-18 under specific circumstances. Current data only support macrophage secretion is one of the sources of IL-18. Authors should draw a safe conclusion of such description.

Response to comment 2:

We absolutely agree with the reviewer and thank a lot for this important comment. We have edited the manuscript to draw safe conclusions and have added new references.

Changes to the manuscript:

- Abstract, page 2 (last sentence)
- Introduction, page 4 (last paragraph)
- Discussion, page 19 (first and second paragraph)
- New references 26, 27

Comment 3:

IL-18 is a complex cytokine. IL-18Ra knock-out mice have been widely used to investigate the function of IL-18. However, recently published work showed that other IL-18R independent mechanisms can elucidate the function of IL-18. In current version of manuscript, authors should pay attention to the IL-18 independent pathways to comprehensively interpret the findings in this manuscript.

Response to comment 3:

We agree with the reviewer and have edited the discussion section of the manuscript to take the IL18-independent pathways into account and have added a new reference.

Changes to the manuscript:

- Discussion, page 19 (last paragraph) and page 20 (first paragraph)
- New reference 29

Comment 4:

Detailed description of IMSA101 is lacked. It is strongly suggested that authors add more info. about this novel sting agonist. It is easy to cite more references including the preclinical data of IMSA101 to the potential readers.

Response to comment 4:

We thank the reviewer for this important comment. As mentioned previously, IMSA101 is a newly developed and previously unpublished STING agonist. US patents covering STING agonists including IMSA101 are filed (10,336,786 and 10,519,188). In order to provide additional functionality data of IMSA101, we extended our investigations and have performed following experiments which are now included in the manuscript:

- *In vivo* anti-tumor efficacy comparison of IMSA101 versus cGAMP (B16 flank tumor model).
- Surface staining and flow cytometry of intratumoral (CAR) T cells from tumor single cell suspension (PDA7940b and B16-huCD19 flank tumor models). This allowed us detailed analyses of IMSA101-mediated effects on intratumoral CART and endogenous T cells (CD4/CD8 ratio, phenotype, activation markers, viability, T_{reg} count).
- Analyses of IMSA101-mediated changes in cytokine levels from mouse tumors (17-plex Luminex; PDA7940b tumor model) and more specifically the time-interval of IMSA101-mediated IL-18-induction in tumors (ELISA; PDA7940b flank tumor model).

Changes to the manuscript:

- Introduction, page 4 (last paragraph)
- Results, pages 5+6, 11, 12+13, and 15-17
- Discussion, page 19 (first paragraph)
- Methods, pages 28-31, 34+35
- New main figures 1, 3h, 4a-e, 5d, and 5h+i
- New supplementary figures 7b, 8, 9, and 11

Reviewer 2:

CART cell therapy demonstrated clinical benefit in the treatment of hematological tumors. However, clinical investigations revealed that this therapeutic venue was ineffective against solid tumors. The present study aims to address this issue and proposes a combination strategy to enhance the efficacy of CART cell therapy against solid tumors. The strengths of the present study rely on the potential clinical relevance and on the *in vivo* results obtained. The results clearly indicate that in the presented settings the combination of a novel STING agonist (IMSA101) and CART cell treatment yields potent anticancer effects. There are unfortunately also several limitations that detract from the quality of the work. The main weakness of the work is the absence of a conceptual advance. The work of Conde et al. (JITC, 2021) showed that the combination of a STING agonist with CART cell therapy triggered anticancer effects. Specific comments follow.

Response:

We wish to thank the reviewer very much for the time and effort, and highly appreciate the comments on the article. We have followed the reviewer's suggestions and changed the manuscript as outlined below.

Comment 1:

The authors show data indicating that IMSA101 has a superior stability compared to cGAMP. While this is certainly of interest, how ISMA101 specifically affects STING signaling is not reported here. As mentioned by the authors, the strength of STING signaling will differentially affect all immune cell subsets both directly and indirectly. What is the effect of ISMA101 on T cells for instance? Will it drive T cell death? Will it enhance T cell activation? Will it favor T cell differentiation into effector or regulatory subsets? These questions have been addressed with cGAMP (please see, for instance, in addition to the work of Conde et al., JITC, 2021, Kuhl et al., EMBO Rep., 2023 and Ni et al., JITC, 2022). In the absence of additional information on IMSA101, and *in vivo* comparisons using cGAMP, it is difficult to assess the potential clinical translation of the presented combination.

Response to comment 1:

We absolutely agree and thank the reviewer for this important comment. As requested by the reviewer, we have performed additional *in vivo* experiments to extend our investigations **i)** to compare anti-tumor efficacy of IMSA101 to cGAMP, and **ii)** to analyze IMSA101-mediated effects on intratumoral CART and endogenous T cells.

- **i)** We have added *in vivo* data comparing anti-tumor efficacy of IMSA101 versus cGAMP (B16 flank tumor model). Mice treated with IMSA101 displayed significantly reduced growth rates when compared to cGAMP at equal doses, which resulted in significantly improved overall survival (Fig. 1). Notably, 40% of mice receiving IMSA101 showed complete remission, while all mice receiving cGAMP at the same doses had to be sacrificed due to tumor progression (Fig. 1).
- **ii)** We had already shown in transcriptomics data of intratumoral CART that i.t. IMSA101 induces intratumoral CART activation and increases expression of immunoregulatory

and/or pro-inflammatory pathways. To analyze IMSA101-mediated effects on intratumoral CART (CD3⁺CD45.1⁺) and endogenous T cells (CD3⁺CD45.1⁻) in more detail, we have repeated animal experiments (PDA7940b and B16-huCD19 flank tumor models) and have harvested tumors for surface staining and flow cytometry to compare T cell subsets (CD4/CD8), T cell activation (PD-1), naïve/memory/effector T cell phenotype (CD44/CD62L), and T cell viability between cohorts. In addition, intracellular staining was performed for T_{reg} counts (CD4⁺CD25⁺FoxP3⁺).

In both syngeneic models, significantly more intratumoral CD8-positive CD3⁺CD45.1⁺ cells could be detected in combination treatment group when compared to CART-alone group. In contrast, levels of CD4-positive CD3⁺CD45.1⁺ were significantly higher in the CART-alone compared to the combination treatment group (Fig. 4b and supplementary fig. 8a), resulting in a significant difference in intratumoral CD4/CD8 cell ratio between IMSA101+CART and CART-alone cohorts (Fig. 4c). Significant differences in CD4/CD8 cell ratio were also observed in intratumoral CD3⁺CD45.1⁻ cells between IMSA101+CART and CART-alone cohorts, but not between combination treatment and IMSA101-alone groups (Supplementary Fig. 9a).

More intratumoral CD4-positive and CD8-positive CD3⁺CD45.1⁺ cells expressed the activation marker PD1 in tumors receiving IMSA101+CART when compared to CART-only in both animal models (Fig. 4d). Further, significantly more CD4-positive or CD8-positive CD3⁺CD45.1⁻ cells expressed PD-1 in the combination treatment group when compared to CART-alone group in the PDA7940b or B16-huCD19 tumor models, respectively (Supplementary Fig. 9b). Also, significantly more intratumoral T cells, either infused or endogenous, showed CD44⁻CD62L⁻ effector cell phenotype in the combination group compared to the CART-alone group (Fig. 4e and Supplementary Fig. 9c). No differences were observed between groups in the rate of dead intratumoral CD3⁺CD45.1⁺ cells in either syngeneic model (Supplementary Fig. 8b and 9d). In contrast, a higher rate of dead CD3⁺CD45.1⁻ cells was seen between the combination treatment group when compared to the CART-alone group in the PDA7940b tumor model (Supplementary Fig. 9e). No difference in rates of dead CD3⁺CD45.1⁻ cells between cohorts were observed in the B16-huCD19 tumor model (Supplementary Fig. 9e). In accordance with previously shown IHC data, we observed significantly lower levels of intratumoral T_{regs} in IMSA101-treated tumors (Figure 3h).

Changes to the manuscript:

- Introduction, page 4 (last paragraph)
- Results, pages 5+6, 11-13
- Discussion, page 19 (first paragraph), 22 (first paragraph)
- Methods, pages 25, 28+29, 30, 34+35
- New main figures 1, 3h, 4a-e
- New supplementary figures 7b, 8, and 9

Comment 2:

The results are compelling and supportive but considering the previous work of Conde et al. (JITC, 2021), it would have been of interest to document why IL-18 is especially important in that context. Is it due to IMSA101 or to the specific settings of the presented experiments?

Response to comment 2:

We thank the reviewer for this comment. As outlined in the discussion section of the manuscript, the anti-tumor effects of IL-18 through activation of lymphocytes is known and has been studied for many years. Effects of IL-18 specifically on CART function has been investigated at our center in detail (references 36,37). Importantly, we have shown that IL-18 significantly enhances the engraftment of human CD8⁺ effector T cells in mice. Further, we developed IL-18 secreting CART which has shown promising anti-tumor responses in first patients treated within a phase I clinical trial.

In this study, we provide evidence that IMSA101 induces IL-18 secretion, as shown in serum cytokine data as well as RNA-ISH from treated tumors from two syngeneic models (PDA7940b and B16-huCD19, Fig. 5). We believe that the IMSA101-mediated IL-18 induction improves CART function, as reflected by *in vivo* comparison of wild type CART with IL18R-negative CART (Fig. 6). To further strengthen the hypothesis that IMSA101 induces IL-18 secretion, we have extended our investigation:

- In addition to already shown serum cytokine data, we repeated the *in vivo* experiment and measured cytokines from mouse tumors (17-plex Luminex; PDA7940b tumor model)
- We analyzed the time-interval of IMSA101-mediated IL-18-induction in tumors (ELISA; PDA7940b flank tumor model).

Indeed, we could detect more IL-18 in mouse tumors 24 hours after i.t. IMSA101 injection and IL-18 levels increased over the observation period of six days (Supplementary Fig. 11b-d). In addition to IL-18, we could detect significantly higher cytokine levels - primarily pro-inflammatory cytokines - in mouse tumors, e.g. IL-1 β , IL-2, IL-5, IL-9, IL-22, IL-23, GM-CSF, IFN γ , and TNF (Figure 5d and Supplementary Fig. 11a). Taken this data and previous literature together, we can conclude that IMSA101 induces secretion of IL-18, which then enhances CART function.

Changes to the manuscript:

- Results, pages 14-16
- Methods, pages 30+31
- New main figure 5d
- New supplementary figure 11a-d

Comment 3:

The authors present insufficient evidence to claim that macrophages are among the major cell types that secrete IL-18 in response to the combination therapy. What is their status? Are DCs involved as well? Is an activation of the inflammasome involved? Additional molecular studies are required.

Response to comment 3:

We thank the reviewer a lot for this important comment. IL-18 is an inflammasome-dependent pro-inflammatory cytokine which requires caspase-1 for processing (caspase-1 is activated by inflammasomes). IL-18 is produced by various cells, including macrophages and DCs. Once activated within inflammasomes, caspase-1 cleaves and processes IL-18, which is then released from the cell. We have edited the discussion section of the manuscript and have extended our analyses regarding IL-18 as requested:

- We have performed additional analyses of our flow cytometry data and determined that the majority of intratumoral macrophages in our analyses showed pro-inflammatory M1 (CD86⁺F4/80⁺MHCII⁺) rather than anti-inflammatory M2-like status (CD86⁺F4/80⁺MHCII⁻CD206⁺) (Supplementary Fig. 5b). To determine if intratumoral macrophages secrete higher levels of IL-18 after IMSA101 treatment, we ascertain the levels of IL18 RNA in DAPI⁺ADGRE1⁺ cells (=macrophages) in IMSA101-treated compared to untreated tumors. We observed that tumors treated with IMSA101 not only exhibit an increase in macrophage number, but also increased IL-18 RNA/macrophage in the TIME (Fig. 5h+i).
- We also analyzed levels of IL-18 mRNA in DCs from formalin-fixed paraffin-embedded tumor tissue using RNA-ISH technology. We observed co-expression of IL18 and ITGAX RNA. However, we did not observe a statistical increase of IL-18 mRNA in the DCs of IMSA101 treated versus untreated tumors (Supplementary Fig. 11f) and there was no difference in the number of DCs in our models.

Changes to the manuscript:

- Results, pages 10 and 16
- Discussion, page 19 (last paragraph)
- Methods, pages 33+34
- New main figure 5i
- New supplementary figures 5b and 11f
- New references 26+27

Comment 4:

Are T cell functions intrinsically affected by the deficiency of IL-18 signaling? As fairly acknowledged by the authors, the contribution of IL-18 is partial in the presented experiments and additional explorations are required.

Response to comment 4:

We absolutely agree with the reviewer and have performed additional *in vitro* assays to compare functionality of IL18R-neg. CART to WT-CART. No differences in T cell expansion and T cell size were observed during manufacturing between IL18R-neg. CART and WT-CART (Supplementary Fig. 12c). Further, no differences in cytokine secretion (TNF, IFN γ , and IL-2) as well as cytotoxicity between IL18R-neg. CART and WT-CART were seen when co-incubated with tumor cells (Fig. 6b+c). Based on this data, we could not observe intrinsically impaired T cell functions due to IL-18 receptor deficiency.

Changes to the manuscript:

- Results, page 17
- Methods, page 26
- New main figures 6b+c
- New supplementary figure 12

Comment 5:

In summary, the authors have demonstrated in two preclinical models that the combination IMSA101 and CART cell therapy is effective against solid tumors. While these results are interesting, the conceptual advance here presented is limited regarding the large number of studies that have previously addressed the relevance of combination therapies using STING ligands.

Response to comment 4:

We thank the reviewer for this comment and agree that there are prior publications on CART and STING agonists (see references 14,15). In this study, we combined CART with a newly developed and previously unpublished STING agonist, IMSA101 which is a cGAMP analogue and which shows enhanced serum stability and superior *in vivo* anti-tumor efficacy (shown now in the manuscript) compared to conventional cGAMP. Therefore, combining IMSA101 with CART represents an attractive and highly translational approach. Further, we have analyzed IMSA101-mediated changes of the TIME; in particular cytokine levels in serum and tumors as well as transcriptomic changes of intratumoral CART including pathway analyses which was not reported in previous publications. We found that enhanced CART anti-tumor function was in part enhanced through IMSA101-mediated IL-18 induction and thus, our study provides new mechanistical insights into how STING agonists improve CART function and extends findings of previous publications.

Reviewer 3:

The authors used tumor bearing syngeneic models to show combination i.t. STING agonist (IMSA101) and CART cells was significantly more effective at controlling tumour growth and improved mouse survival, than CART or IMSA101 alone. The responding mice were resistant to re-challenge and had evidence of epitope spreading. Increased tumor-infiltrating endogenous immune subsets and CAR-T cells were present in the PDA7940b (but not B16-huCD19) tumors when co-treated with IMSA101. Combination CART + i.t. IMSA101 treatment showed significantly increased expression of genes in T cell activation and cytokine signalling (IL18 pathway highest). Mouse serum IL18 was significantly higher in the cohort treated with CART + i.t. IMSA101 in both models, transcript for IL18 and ADGRE1 (CD68) were co-located. IL18R KO CAR-T cells were less effective than WT CAR-T cells at controlling B16-huCD19 tumour growth, when used in combination with i.t. IMSA101.

Response:

We highly appreciate the reviewers' comments and suggestions and wish to thank the reviewer for the time and effort. We have followed the reviewer's suggestions and changed the manuscript as outlined below.

Comment 1:

Figure 2G. The IHC images show a differential distribution of LY6G⁺ and F4/80⁺ cells in treated tumors. Are these images from serial sections of the same tumors? Are all combination treated tumors characterized by excluded F4/80⁺ cells?

Response to comment 1:

We thank the reviewer a lot for this comment and have edited the figure to make sure that representative images for both markers, Ly6G and F4/80 are from same tumors. Over 10 sections were obtained from the paraffin blocks of the tumors for various histochemical and immunohistochemical stains, which makes some tumor sections appear differently across the different sections (e.g., sections for IMSA101-only in figure 2G). This information was added to the methods section of the manuscript.

Representative IHC images confirm infiltration of F4/80⁺ cells into the tumors of IMSA101-alone and combination treatment groups, while F4/80⁺ cells were primarily detected at tumor edges without notable infiltration in untreated and CART-alone cohorts (see Fig. 3g). Thus, combination treated tumors are characterized with F4/80⁺ cell infiltration rather than exclusion. This is also in accordance with flow cytometry (Fig. 2e), IHC (Fig. 2f), and RNA-ISH data (Fig. 5g), where significantly increased counts of macrophages were detected in IMSA101-treated tumors.

Changes to the manuscript:

- Results, page 11
- Methods, pages 32 (last paragraph) and 33 (first paragraph)
- New main figure 3g

Comment 2:

Figure 3 shows nanostring GEP data for [IMSA101 + CART] vs CART alone. This is very interesting and shows the combination treatment induces increased expression of genes for T cell immune checkpoint molecules, pro-inflammatory chemokines, IFNA1 and gzmB. However, IFNg does not appear to be differentially expressed between these two treatment cohorts, and suggests the IFN-g response pathway is not critical for the increased anti-tumor effect. Can the authors provide data which explains the trafficking signals necessary for the CAR-T cell and endogenous immune cell trafficking into these tumors?

Response to comment 2:

We thank the reviewer for this important comment. IFNg is known to be needed for CART killing in solid tumors (DOI: 10.1038/s41586-022-04585-5). We had already shown increased IFNg levels in serum of mice receiving IMSA101+CART when compared to CART-alone (Fig. 5c). To deepen our analyses, we have performed additional *in vivo* experiments to analyze IFNg levels in mouse PDA7940b tumors. Therefore, we harvested tumors at day 6 after treatment for 17-plex Luminex assay. In accordance with already shown serum cytokine data, levels of intratumoral IFNg was significantly higher in mice receiving IMSA101+CART when compared to CART only (Fig. 5d and Supplementary Fig. 11a). This supports the hypothesis that IMSA101-mediated changes within the TIME induces IFNg secretion, which is required for CART killing.

Changes to the manuscript:

- Results, page 15
- Methods, pages 30+31
- New main figure 5d
- New supplementary figure 11a

Comment 3:

Figure 4 shows combination treatment induced IL18 production in macrophages in the tumor. Can the authors provide quantitative data to show the % and/or number of macrophages that make IL18 compared to untreated or monotherapy tumors? What is the time interval for the increased IL18 production?

Also the combination drug-treated mice have increased IL18 in the serum (B-C). Can the authors also provide data exploring the same cytokines in the tumor?

Response to comment 3:

We thank the reviewer for this comment and have performed additional *in vitro* and *in vivo* experiments to address this important comment as follows:

- As requested by the reviewer, Dr. Charles-Antoine Assenmacher, veterinary pathologist and co-author on the paper, developed an algorithm using RNA-ISH to determine the rate of intratumoral macrophages secreting IL-18. We could detect significantly higher levels of IL18 RNA in DAPI⁺ADGRE1⁺ cells (=macrophages) in IMSA101-treated tumors, suggesting that IMSA101 not only increases macrophage count in the TIME, but also increases the rate of IL18-secreting intratumoral macrophages (Fig. 5i).

- As requested by the reviewer, we performed another *in vivo* experiment to determine the time interval of IMSA101-mediated IL18 induction. Therefore, IMSA101-treated tumors were harvested at the following time points after i.t. IMSA101 administration: 4 hours, 24 hours (1 day), 72 hours (3 days), and 144 hours (6 days). Untreated tumors were used for control and IL-18 levels from tumors were measured using ELISA. Already 4 hours after i.t. IMSA101 injection, an increase of IL-18 levels in IMSA101-treated tumors was observed when compared to untreated tumors, reaching a significant difference at 24 hours after IMSA101 treatment. A further increase of IL-18 levels was seen in tumors over the observed period of 6 days (Supplementary Fig. 11b-d).
- We performed another *in vivo* experiment to analyze levels of a total of 17 cytokines in treated tumors for correlation with already shown serum cytokine levels, as requested by the reviewer. Therefore, tumors were harvested 6 days after treatment and tumor cytokines were analyzed using 17-plex Luminex assay. Significantly higher levels of IL-1 β , IL-2, IL-5, IL-9, IL-18, IL-22, IL23, GM-CSF, IFN γ , and TNF were detected in tumors of mice receiving combination treatment when compared to mice receiving CART alone. In contrast, significantly lower IL-10 levels were observed in the combination treatment group when compared to the CART-only group (Fig. 5d and Supplementary Fig. 11a).

Changes to the manuscript:

- Results, pages 14-16
- Methods, pages 30+31, and 33
- New main figures 5d+i
- New supplementary figure 11

Comment 4:

Figure 5 indicates that IL18 response by CAR-T cells plays a role in the anti-tumor efficacy of combination i.t. IMSA101 + CAR-T therapy. There are a series of questions that emanate from this finding:

1. What effect does IL-18 have on the functional polarisation of the tumour infiltrating CAR-T cells? Is there a difference in CD4⁺ vs CD8⁺ CAR-T cell response to IL18?
2. Which endogenous immune cell types in the tumor express IL18R? What effect does IL18 signaling have on their functional responses in the tumour?
3. A number of cytokine signalling pathways are induced by IMSA101 + CAR-T therapy in the tumor (Fig 5). These may also play a critical role in the anti-tumour efficacy of this combination treatment. This could either be in concert with IL18 or as a parallel driver of efficacy. Can the authors provide data to show which of the upregulated cytokine pathways are critical for the anti-tumour effect?
4. What time interval is critical for IMSA101-induced IL-18 production in the TME to induce the anti-tumor effect?

Response to comment 4:

We thank the reviewer and have addressed the comments as follows:

- **1.** As outlined in the discussion section of the manuscript, the anti-tumor effects of IL-18 through activation of lymphocytes is known. IL-18 affects both, CD4⁺ and CD8⁺ T cells: While it enhances cytotoxic activity of CD8⁺ T cells, it stimulates the production of IFN γ in CD4⁺ and CD8⁺ T cells. Effects of IL-18 specifically on CART function has been investigated at our center in detail (references 36,37). Importantly, we have shown that IL-18 significantly enhances the engraftment of human CD8⁺ effector T cells in mice. Further, we developed IL-18 secreting CART which has shown promising anti-tumor response in a phase I clinical trial.

Given that IMSA101 also induces IL18-secretion, we analyzed IMSA101-mediated effects on intratumoral CART and endogenous T cells in more detail. We had already shown in transcriptomics data of intratumoral CART that i.t. IMSA101 induces intratumoral CART activation and increases expression of immunoregulatory and/or pro-inflammatory pathways (Fig. 4f-i). To analyze IMSA101-mediated effects on intratumoral CART (CD3⁺CD45.1⁺) and endogenous T cells (CD3⁺CD45.1⁻) in more detail, we have repeated animal experiments (PDA7940b and B16-huCD19 flank tumor models) and have harvested tumors for surface staining and flow cytometry to compare T cell subsets (CD4/CD8), T cell activation (PD-1), naïve/memory/effector T cell phenotype (CD44/CD62L), and T cell viability between cohorts. In addition, intracellular staining was performed for T_{reg} counts (CD4⁺CD25⁺FoxP3⁺).

In both syngeneic models, significantly more intratumoral CD8-positive CD3⁺CD45.1⁺ cells could be detected in combination treatment group when compared to CART-alone group. In contrast, levels of CD4-positive CD3⁺CD45.1⁺ were significantly higher in the CART-alone compared to the combination treatment group (Fig. 4b and supplementary fig. 8a), resulting in a significant difference in intratumoral CD4/CD8 cell ratio between IMSA101+CART and CART-alone cohorts (Fig. 4c). Significant differences in CD4/CD8 cell ratio were also observed in intratumoral CD3⁺CD45.1⁻ cells between IMSA101+CART and CART-alone cohorts, but not between combination treatment and IMSA101-alone groups (Supplementary Fig. 9a).

More intratumoral CD4-positive and CD8-positive CD3⁺CD45.1⁺ cells expressed the activation marker PD1 in tumors receiving IMSA101+CART when compared to CART-only in both animal models (Fig. 4d). Further, significantly more CD4-positive or CD8-positive CD3⁺CD45.1⁻ cells expressed PD-1 in the combination treatment group when compared to CART-alone group in the PDA7940b or B16-huCD19 tumor models, respectively (Supplementary Fig. 9b). Also, significantly more intratumoral T cells, either infused or endogenous, showed CD44⁻CD62L⁻ effector cell phenotype in the combination group compared to the CART-alone group (Fig. 4e and Supplementary Fig. 9c). No differences were observed between groups in the rate of dead intratumoral CD3⁺CD45.1⁺ cells in either syngeneic model (Supplementary Fig. 8b and 9d). In contrast, a higher rate of dead CD3⁺CD45.1⁻ cells was seen between the combination treatment group when compared to the CART-alone group in the PDA7940b tumor model (Supplementary Fig. 9e). No difference in rates of dead CD3⁺CD45.1⁻ cells between cohorts were observed in

the B16-huCD19 tumor model (Supplementary Fig. 9e). In accordance with previously shown IHC data, we observed significantly lower levels of intratumoral T_{regs} in IMSA101-treated tumors (Figure 3h).

- **2.** Besides T cells, IL-18R is typically expressed on other immune cells which play important functional roles within the TIME, including NK cells, NKT cells, macrophages, DCs, and B cells. In the presence of IL-12, IL-18 activates and induces the production of IFN- γ by Th1 cells, macrophages, NK cells, NKT cells, B cells, and DCs. In the absence of IL-12, IL-18 with IL-2 induces type 2 T helper (Th2) cytokines from NK cells and NKT cells. Additionally, IL-18, in synergy with IL-3, induces basophils and mast cells to produce IL-4 and IL-13. We have edited the discussion section of the manuscript and added new references.
- **3.** Besides the IL-18 pathway, upregulated pathways in intratumoral CART of the combination treatment group included IL-21, IL-9, IL-2, IL-15, IL-6, and IL-23 (Fig. 4i). Consistent with the transcriptomics data, significantly more IL-18, IL-9, IL-2, IL-23, and other pro-inflammatory cytokines were detected in IMSA101-treated tumors (Supplementary Fig. 11a). IL-21 influences T cell differentiation and can enhance the development of cytotoxic CD8⁺ T cells. As already outlined in the discussion section of the manuscript, we have previously shown that IL-9 signaling can help maintain T cell stemness (reference 39). IL-2 and IL-15 are known to promote stimulation, proliferation, and survival of T cells. IL-6 promotes T cell survival and limits apoptosis, while IL-23 activates Th17 cells. Overall, all upregulated cytokine receptor pathways could be in part responsible for the observed IMSA101-mediated effects on CART. This is in accordance with our observation, that the use of IL18R-neg. CART significantly impaired but did not abrogate anti-tumor response in mice receiving combination treatment, supporting that IMSA101 enhances CART response through IL18-mediated as well as independent mechanisms. We have edited the discussion section of the manuscript and have added new references.
- **4.** As requested by the reviewer, we performed another *in vivo* experiment to determine the time interval of IMSA101-mediated IL18 induction. Therefore, IMSA101-treated tumors were harvested at the following time points after i.t. IMSA101 administration: 4 hours, 24 hours (1 day), 72 hours (3 days), and 144 hours (6 days). Untreated tumors were used for control and IL-18 levels from tumors were measured using ELISA. Already at 4 hours after i.t. IMSA101 injection, an increase of IL-18 levels in IMSA101-treated tumors was observed when compared to untreated tumors, reaching a significant difference at 24 hours after IMSA101 treatment. A further increase of IL-18 levels was seen in tumors over the observed period of 6 days (Supplementary Fig. 11b-d).

Changes to the manuscript:

- Results, pages 11-13, and 15+16
- Discussion, pages 20 (first paragraph) and 21 (second paragraph)
- Methods, pages 30+31, and 34+35
- New main figures 3h, 4a-e

- New supplementary figures 7b, 8, 9, and 11b-d
- New references 26+27, 29+30, and 41

Comment 5:

For the *in vivo* studies, please indicate in the legends whether the data is representative or pooled and from how many experiments.

Response to comment 5:

We thank the reviewer for this comment, and as requested we have edited the figure legends wherever indicated.

Changes to the manuscript:

- See main figure legends 3, 4, and 5
- See supplementary figure legends 5, 6, 9, 11

Comment 6:

Fig 2F, can the authors please explain why the units for F4/80⁺ and LY6G⁺ cells are pixels/mm² rather than cells/ mm².

Response to comment 6:

We thank the reviewer for this comment. As already stated in the methods section of the manuscript (page 32), nuclear algorithms for cell counting were generated for the CD3, FoxP3 and Klrblc/CD161c (NK) stains, and positive pixel count algorithms for area quantification were generated for the Ly-6G and F4/80 stains. The staining properties of the two stains F4/80 and Ly6G made the quantification of individual cells inaccurate and unreliable which is the reason why Dr. Charles-Antoine Assenmacher, veterinary pathologist and co-author in this publication, decided to use the positive pixel count algorithm which very precisely quantifies the area of positivity within the tumor sections for each stain. We have added this additional information into the methods section of the manuscript.

Changes to the manuscript:

- Methods, page 32

REVIEWERS' COMMENTS

Reviewer #1 (Remarks to the Author):

The manuscript by Uslu et al. has been greatly improved, only minor revisions are needed before it can be published.

1. Abbreviation should be written consistently, such as CART, not CAR T.
2. Line 505, 520: "anti-Fab(2)" should be "anti- F(ab')₂".
3. Line 567: "2e+6 AsPC-1 cells in a total volume of 100ul 1:1 matrigel (Corning): 1X DPBS (Gibco) mix were implanted subcutaneously into the right flank of NSG mice" should be polished.
4. Line 608: In the method part, the calculation formulas of tumor volumes are different for the same tumor model (B16-F10). Please explain.
5. Figure 4e: The number of experimental samples for each group should be stated.
6. Figure 3c, g, and figure 5e, h: The scale of the picture should be marked.
7. Figure 6c: Lack of statistical analysis.

Reviewer #2 (Remarks to the Author):

My concerns have been addressed in this revised version of the work.

Reviewer #3 (Remarks to the Author):

The authors have submitted a revised manuscript which addressed the concerns I raised on the original manuscript. They provided new experimental data, analysis and discussion to this revised comprehensive manuscript.

I have no further comments.

Reviewer 1:

The manuscript by Uslu et al. has been greatly improved, only minor revisions are needed before it can be published.

Response:

We highly appreciate the reviewers' comments and suggestions and wish to thank the reviewer for the time and effort.

Comments 1-7:

1. Abbreviation should be written consistently, such as CART, not CAR T.
2. Line 505, 520: "anti-Fab(2)" should be "anti- F(ab')₂".
3. Line 567: "2e+6 AsPC-1 cells in a total volume of 100ul 1:1 matrigel (Corning): 1X DPBS (Gibco) mix were implanted subcutaneously into the right flank of NSG mice" should be polished.
4. Line 608: In the method part, the calculation formulas of tumor volumes are different for the same tumor model (B16-F10). Please explain.
5. Figure 4e: The number of experimental samples for each group should be stated.
6. Figure 3c, g, and figure 5e, h: The scale of the picture should be marked.
7. Figure 6 c: Lack of statistical analysis.

Response to comment 1:

We thank the reviewer for the comments. We have edited the manuscript accordingly. Changes to the text were highlighted in yellow in the revised manuscript.

Reviewers 2 and 3:

Reviewer 2:

My concerns have been addressed in this revised version of the work.

Reviewer 3:

The authors have submitted a revised manuscript which addressed the concerns I raised on the original manuscript. They provided new experimental data, analysis, and discussion to this revised comprehensive manuscript. I have no further comments.

Response:

We wish to thank reviewer 2 and 3 again for their constructive comments and suggestions, which we believe have greatly enhanced the manuscript.